# Learning Interpretable Models Using Uncertainty Oracles

## Abstract

A desirable property of interpretable models is small size, so that they are easily understandable by humans. This leads to the following challenges: (a) small sizes typically lead to diminished accuracy, and, (b) different techniques offer bespoke levers, e.g., L1 regularization, for making this size-accuracy trade-off that might be insufficient to reach the desired balance.

We address these challenges here. Earlier work has shown that learning the training distribution creates accurate small models. Our contribution is a new technique that exploits this idea. The training distribution is modeled as a Dirichlet Process for flexibility in representation. Its parameters are learned using Bayesian Optimization; a design choice that makes the technique applicable to non-differentiable loss functions. To avoid challenges with high data dimensionality, the data is first projected down to one-dimension using uncertainty scores of a separate probabilistic model, that we refer to as the uncertainty oracle.

Based on exhaustive experiments we show that this technique possesses multiple merits: (1) it significantly enhances small model accuracies, (2) is versatile: it may be applied to different model families with varying notions of size, e.g., depth of a decision tree, non-zero coefficients in a linear model, simultaneously the maximum depth of a tree and number of trees in Gradient Boosted Models, (3) is practically convenient because it needs only one hyperparameter to be set and works with non-differentiable losses, (4) works across different feature spaces between the uncertainty oracle and the interpretable model, e.g., a Gated Recurrent Unit trained using character sequences may be used as an oracle for a Decision Tree that uses character n-grams, and, (5) may augment the accuracies of fairly old techniques to be competitive with recent task-specialized techniques, e.g., CART Decision Tree (1984) vs Iterative Mistake Minimization (2020), on the task of cluster explanation.

## 1 Introduction

In recent years, Machine Learning (ML) models have become increasingly pervasive in various real world systems. This has led to a growing emphasis on models to be *understandable*, especially in high human-impact domains, e.g., medicine and healthcare (Caruana et al., 2015; Mienye et al., 2024), defence applications (Gunning, 2016; Moustafa et al., 2023), law enforcement (Angwin et al., 2016; Hall et al., 2022; Herrewijnen et al., 2024).

An important aspect of model interpretability is its size (smaller is better); this has been established through user studies (Feldman, 2000; Kulesza et al., 2013; Piltaver et al., 2016; Lage et al., 2019; Poursabzi-Sangdeh et al., 2021), and is also evidenced by its popularity as an algorithm design criteria (Tibshirani, 1996; Ribeiro et al., 2016; Herman, 2017; Lipton, 2018; Murdoch et al., 2019; Lakkaraju et al., 2016; Good et al., 2023). However, smaller sizes typically imply relatively lower capacity and thus, lower accuracy. A practitioner may control this size-accuracy trade-off using bespoke levers offered by a training algorithm, e.g., early stopping in Decision Trees (DT), L1 regularization in linear models. However, this presents certain challenges: (1) one needs to be intimately aware of how various hyperparameters (hence referred to as *hyperparams*) interact, and (2) the desired trade-off might not even be achievable within its hyperparam search space.

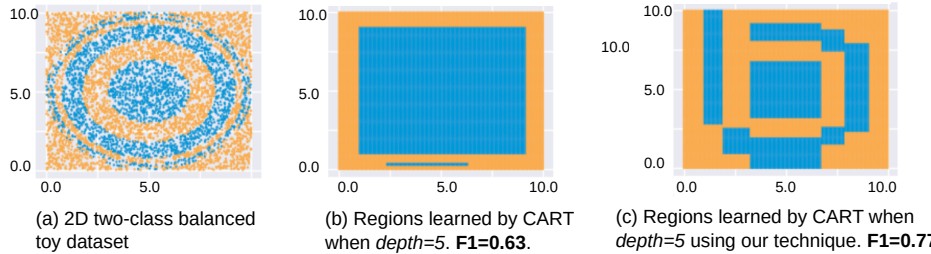

(a) 2D two-class balanced toy dataset

(b) Regions learned by CART when *depth=5*. **F1=0.63**.

(c) Regions learned by CART when *depth=5* using our technique. **F1=0.77**.

Figure 1: Application of our technique is shown on the toy dataset in (a). Learning a DT *constrained* to a depth of 5 using the CART (Breiman et al., 1984) algorithm produces the regions shown in (b). Additionally learning the training distribution using our technique produces the regions in (c). For both (b) and (c) the F1-macro scores on a held-out set are reported.

Here we propose a model-agnostic[1] technique that often produces better accuracies for small-sized models on classification problems. The underlying strategy is to learn a distribution over training instances, that represents their informational value for learning, and sample a new training set accordingly; models thus constructed have been shown to possess favorable size-accuracy trade-offs (Ghose & Ravindran, 2020). Our technique is an implementation of this principle.

The distribution used is a mixture model based on the *Dirichlet Process* - picked for its flexibility of representation and maturity within the Bayesian nonparametrics community. Its parameters are learned using *Bayesian Optimization*, so as to accommodate non-differentiable losses, e.g., many DT and rules learners. To make this process computationally efficient, we avoid directly learning the distribution over the input space which may have high dimensionality. Instead we first project instances down to a single dimension, using an auxiliary model's prediction uncertainty scores. We refer to this model as the *uncertainty oracle*.

As an illustration, consider the toy dataset in Figure 1(a). Figure 1(b) visualizes class regions learned by a DT of $depth = 5$ using the CART (Breiman et al., 1984) algorithm. The F1-macro score on a held-out set is $0.63$. When the training distribution is also learned using our technique, we obtain the regions in Figure 1(c) and a F1-macro score of $0.77$, for the same tree depth. The oracle used is a Gradient Boosted Model (GBM) Friedman (2001).

**Our primary contribution** is a model-agnostic technique that produces small accurate models. It is also agnostic to the notion of model *size*, e.g., number of terms of with non-zero coefficients in a linear model or depth of a DT, *both* the number of trees and depth per tree in GBMs. We show that this produces relative improvements of $\sim 100\%$ in some cases. It is convenient to use as it works with with non-differentiable losses, and only one hyperparam needs to be set.

**Additionally**, we show that: (1) it is more accurate than its predecessor (Ghose & Ravindran, 2020), (2) it can elevate the performance of fairly old techniques to be competitive with relatively new ones, and (3) can use an uncertainty oracle that is trained on a different feature space than what the target model uses. The last property allows for a broad choice of oracles, e.g., in the case of text classification, the oracle might be a *Gated Recurrent Unit (GRU)* that is learned using a sequence of characters while the target model might be a DT over *n-grams*.

□

The rest of the paper is organized as follows: we first review related work in §2. We then detail our technique in §3. We follow that up with rigorous empirical validations in §4.A side effect of allowing non-differentiable losses is high running times; we discuss this limitation, and a mitigation in §5. Finally, we conclude with a discussion on future work in §6.

---

[1]We use the term to mean agnostic to the model *family*, as is accepted usage in the area of XAI, e.g., SHAP (Lundberg & Lee, 2017), LIME (Ribeiro et al., 2016).

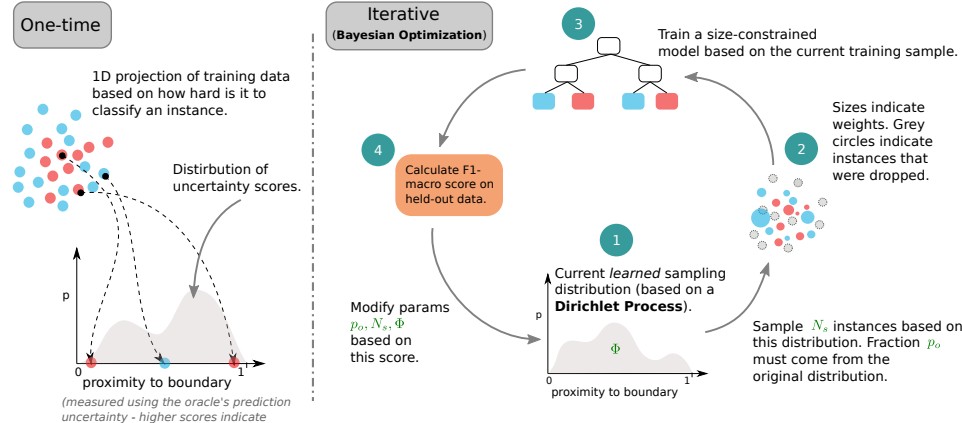

Figure 2: Overview of our technique. *Left*: Training instances are characterized by their proximity to class boundaries. As a proxy for this quantity, we use the prediction uncertainty scores of a prob-abilistic oracle (these may also be seen as an 1D *projection*): higher uncertainty indicates proximity to a boundary. These scores are calculated *once*. *Right*: The size-constrained model is learned iteratively. A sampling distribution, parameterized by $\Phi$, over the uncertainty values (shown in `Step 1`) is used to sample training instances (as in `Step 2`), which is used to train a size-constrained model (shown in `Step 3`). Its accuracy on a held-out set - `Step 4` - is used to modify $\Phi$. This loop, `Steps 1-4`, is executed by a BayesOpt algorithm.

## 2 RELATED WORK

The concept of using a different training distribution relative to test is common in the case of class imbalance, e.g., undersample the majority class data (Japkowicz & Stephen, 2002; Chawla et al., 2002; He et al., 2008; Santhiappan et al., 2018), but it was shown to be a general strategy for improving accuracy in Ghose & Ravindran (2020). Their technique relies on a specialized DT, called *density tree*, that encodes the geometric placement of training data. We believe that using these trees - which are primarily learned using the CART algorithm - inherently limits the accuracy of their technique. This work may be seen as a non-trivial extension: since the uncertainty oracle can come from an arbitrary model family, it provides greater flexibility and accuracy.

The interaction of two models - the oracle and the interpretable model - suggests an overlap with the area of *Knowledge Distillation* Gou et al. (2021). But there is a critical difference: *in theory, we don't require the oracle model*; here it happens to be a convenient *tool for dimensionality reduction*. Indeed, there are other ways to achieve a similar outcome, e.g., within *Active Learning*, it is common to infer proximity to a class boundary by noting the labels of an instance's neighbors (Margatina et al., 2021; Chen et al., 2023); these setups might be thought of as rudimentary *k-Nearest Neighbor (kNN)* models. With our use of the oracle, we avoid having to worry about neighborhood-related hyperparams, such as neighbor distance.This should not be seen as distillation for the same reason as we don't consider using such *kNN*s as effecting distillation. The oracle's peripheral role is also underscored by the fact that its labels are ignored. This lack of fidelity wrt the oracle is also why our technique shouldn't be seen an explanation technique, i.e., *XAI*, such as *TREPAN* (Craven & Shavlik, 1995) or *LIME* (Ribeiro et al., 2016).

## 3 METHODOLOGY

We begin describing our technique with an overview. This is then used as a foundation for introducing details.

### 3.1 OVERVIEW

Our technique is visualized in Figure 2. Instead of learning the training distribution directly, which might be expensive because of the dimensionality of the data, we first project the data down to *one*

dimension. This is done just once, and is shown in the left panel in Figure 2. Since we are solving for classification, we want this dimension to correspond to the "classifiability" of an in instance, or how close is an instance to a class boundary. As a tractable proxy for this property, we train a separate highly accurate probabilistic *oracle* model on the training data[2], and use its prediction *uncertainty* score as the projected value; high uncertainty scores typically denote proximity to class boundaries Lewis & Catlett (1994).

The distribution is modeled as an *Infinite Beta Mixture Model* using a *Dirichlet Process*, which is *iteratively* learned. `Step 1` on the right panel in Figure 2 shows the current distribution, based on which training data is sampled (`Step 2`). The size-constrained model of interest is then trained on this sample - `Step 3` - and its accuracy on a held-out set is calculated - `Step 4`. This score is used as a feedback for the optimizer which repeats the process to learn better distribution parameters. We use *Bayesian Optimization (BayesOpt)* (Shahriari et al., 2016; Garnett, 2023) to accommodate models with *non-differentiable* loss functions. Note that we can't just pick highly uncertain points, because that has not been shown to consistently work well (Ghose & Nguyen, 2024).

## 3.2 TERMINOLOGY AND NOTATION

We introduce some nomenclature before discussing our algorithm.

1. A dataset is denoted as a set of instance-label pairs, $D = \{(x_1, y_1), (x_2, y_2), ..., (x_N, y_N)\}$. A joint distribution over a dataset is denoted by $p(X, Y)$.

2. To distinguish between the distribution we are given (in form of the dataset) and the one we learn, we refer to the former as the *original* distribution. In *all* experiments here, the test and held-out data follow the original distribution; for the training data, we learn a new distribution.

3. We let $acc(M, p)$ denote some classification accuracy metric for model $M$ on data represented $p(X, Y)$.

4. $train_{\mathcal{F}, f}(p, \eta)$ is understood to produce a model of size $\eta$ (for some pre-decided notion of size) from the model family $\mathcal{F}$ using a specific training algorithm $f$.

   For instance, $\mathcal{F}$ might represent DTs and $f$ might be the CART algorithm, and $\eta = 5$ might denote a DT of $depth = 5$. We let $\eta = *$ denote unbounded size.

Let us state our objective using this notation. Typically, a model is trained on the same distribution as the test (on which it is evaluated), i.e., we evaluate $acc(train_{\mathcal{F}, f}(p, \eta), p)$. In this work, the training distribution is allowed to be different relative to the test. In other words, we seek $p'$ such that:

$$\underset{p'}{\arg\max}\, acc(train_{\mathcal{F}, f}(p', \eta), p) \tag{1}$$

## 3.3 ALGORITHM

Referring to the high-level flow in Figure 2, we note that the proposed technique relies on a few important ingredients. These are described below, while a more comprehensive discussion may be found in §A.4:

1. **Uncertainty score**: This is needed for the one-time projection using the oracle. There are multiple ways to measure prediction uncertainty; here we choose *margin uncertainty* (Scheffer et al., 2001), since (a) it accounts for prediction probabilities of different classes, (b) while also producing high scores even with two dominant predicted classes in a setting with more classes. The uncertainty score for $x$, as provided by model $M$, is denoted by $u_M(x) \in [0, 1]$. The margin uncertainty is calculated as:

$$u_M(x) \leftarrow 1 - (p_{C_1} - p_{C_2}) \tag{2}$$

   Here, $p_{C_1}$ and $p_{C_2}$ denote the probabilities of the most confident and next most confident classes. See §A.1 for further details.

---

[2]Performed using cross-validation or using a random held-out set, to avoid overfitting.

2. **Density model**: Since we want to *learn* a distribution, we want the representation to be flexible. We encode the density as a *mixture model* of *Beta* distributions. We use the latter since (a) their support matches the range of uncertainty scores, i.e., $u_M(x) \in [0,1]$, and (b) a *Beta mixture model* can approximate any distribution in $[0,1]$ arbitrarily well (Diaconis & Ylvisaker, 1983). Further, in the interest of flexibility, we refrain from explicitly dictating the number of *Beta* components, and thus, we use an *Infinite* Beta Mixture Model (IBMM), where the component assignments are decided by a standard *Dirichlet Process (DP)* (Ferguson, 1973). This is a popular tool in the area of *Bayesian Nonparametrics* Yee Whye Teh & Blei (2006); Wang et al. (2011). Another advantage of this formulation is that it leads to a fixed number of parameters irrespective of the number of active components which makes it easy to pick an optimizer. We note here that Ghose & Ravindran (2020) also use a DP-based IBMM, but for modeling the height of density trees.

   Two sets of parameters are required to describe this density model:

   (a) The shape parameters $A_i, B_i$ of the $i^{th}$ *Beta* component. These are separately sampled from prior distributions that are themselves *Beta* distributions, with shape parameters $a, b$ and $a', b'$ respectively. Since naively doing this would restrict $A_i$ or $B_i$ to *Beta's* support, i.e., $[0,1]$, we also multiply the sampled value by a variable *scale*, that we set to be large enough to cover the family of component distributions we require[3]. Effectively then, $A_i \sim scale \times Beta(a,b)$ and $B_i \sim scale \times Beta(a',b')$.

   (b) The DP needs just a *concentration* parameter $\alpha \in \mathbb{R}_{>0}$ that decides the number of active components, i.e., ones with instances assigned to them[4].

   In all, the density model requires *five* parameters, which we denote as $\Psi = \{\alpha, a, b, a', b'\}$. To sample $N_s$ instances given $\Psi$, we first determine the number of instances per component using a standard technique like the *Chinese Restaurant Process* (Aldous, 1985) and then sample component-wise. Please see §A.2 for details.

3. **Optimization**: As mentioned earlier, we use BayesOpt to accommodate non-differentiable losses. It is also resilient to noise, which is relevant due to factors such as randomized initialization of model parameters, different dataset splits across trials, etc. Specifically, we use the *hyperopt* library (Bergstra et al., 2013), which implements the *Tree Structured Parzen Estimator (TPE)* algorithm Bergstra et al. (2011). *Because there is no tight coupling between our formulation and the optimizer, it is possible to use a different BayesOpt library.* This can be a crucial practical consideration, and is discussed in §5.

   For optimization, in addition to $\Psi$, we retain the following parameters, originally introduced in Ghose & Ravindran (2020):

   (a) $N_s$: This is the sample size - this is also learned.

   (b) $p_o \in [0,1]$: Proportion of the new training set that is uniformly sampled from the original training data. This serves two purposes: (1) it acts as a "shortcut" for the optimizer to mix in the original distribution as needed, and (2) it serves as a "probe variable", i.e., it shows how much of the original distribution is actually needed for good accuracies.

   Accounting for these, we now have a total of *seven* optimization variables: $\Psi = \{\alpha, a, b, a', b'\}, N_s, p_o$, which are iteratively optimized, till the **budgeted number of iterations**, $T$, are exhausted. These variables are collectively denoted as $\Phi = \{\Psi, N_s, p_o\}$. Algorithm 1 outlines the overall technique; here the interpretable and oracle model families are denoted by $\mathcal{I}$ and $\mathcal{O}$, and the respective training algorithms are denoted by $h$ and $g$ respectively. §A.4 provides additional details around model selection, robust estimation of $acc$, etc.

**Optimization variables and parameters**: The task of the optimizer is to find $\Phi$ that maximizes the held-out accuracy (line 11 in Algorithm 1) within $T$ iterations. The optimizer here accepts *box constraints*, and as such their lower/upper bounds, which need to be set by the user, are *parameters* (along with $T$) of the technique. We discuss in §A.3 that reasonable default bounds exist for parameters $\Phi$, e.g., its easy to see $p_o \in [0,1]$. So, **in practice, $T$ is the only parameter that a user needs to set**.

---

[3]**NOTE**: This is fixed at a value of 10000 and not learned; hence it isn't counted as a parameter.

[4]Of course, in theory, there are an infinite number of components, but the number of *active* components grows with data.

---

**Algorithm 1:** Learning interpretable model using oracle

---

**Data:** Dataset $D$, model size $\eta$, $train_{\mathcal{O},h}()$, $train_{\mathcal{I},g}()$, iterations $T$

**Result:** Optimal parameters $\Phi^*$, test set accuracy $s_{test}$ at $\Phi^*$, and interpretable model $M^*$ at $\Phi^*$

1 Create splits $D_{train}, D_{val}, D_{test}$ from $D$, stratified wrt labels. Here
   $|D_{train}| : |D_{val}| : |D_{test}| :: 60 : 20 : 20$.

2 $M_O \leftarrow train_{\mathcal{O},h}(D_{train}, *)$

3 **for** $t \leftarrow 1$ **to** $T$ **do**

4     $\Phi_t \leftarrow suggest(s_0, s_1, ...s_{t-1}, \Phi_0, \Phi_1, ..., \Phi_{t-1})$ `// ` $s_0, \Phi_0$ ` initialized at ` $t=0$`,`
         `see text.   Note: ` $\Phi_t = \{\Psi_t, N_{s,t}, p_{o,t}\}$ ` where ` $\Psi_t = \{\alpha_t, a_t, b_t, a'_t, b'_t\}$`.`

5     $N_o \leftarrow p_{o,t} \times N_{s\_t}$

6     $N_u \leftarrow N_{s\_t} - N_o$

7     $D_o \leftarrow$ uniformly sample with replacement $N_o$ points from $D_{train}$

8     $D_u \leftarrow$ sample $N_u$ points from $D_{train}$ using the DP-based IBMM given current values for
         $N_u, M_O, D_{train}, \Psi_t$ `// see Algorithm A.2 for details`

9     $D_s \leftarrow D_o \uplus D_u$ `// ` $D_o$`, ` $D_u$ ` are assumed to be multisets`

10     $M_t \leftarrow train_{\mathcal{I},g}(D_s, \eta)$

11     $s_t \leftarrow acc(M_t, D_{val})$

12 **end**

13 $t^* \leftarrow \arg\max_t \{s_1, s_2, ..., s_{T-1}, s_T\}$

14 $\Phi^* \leftarrow \Phi_{t^*}$

15 $M^* \leftarrow M_{t^*}$

16 $s_{test} \leftarrow acc(M^*, D_{test})$

17 **return** $\Phi^*, s_{test}, M^*$

---

**Smoothing**: A final practical consideration is the smoothness of the optimization landscape. Uncertainty scores over the training data may often result in a density that isn't smooth, making it difficult to learn a good distribution. We redress this by explicitly smoothing the density. We detail this in §A.5.

This concludes our discussion of algorithmic details; next, we look at empirical validation.

## 4 EXPERIMENTS

We have performed extensive empirical investigations to validate the utility of our technique. These may be grouped in the following manner:

1. Those that establish the *effectiveness* of the technique in various settings, i.e., different datasets, interpretable models and oracles, across different model sizes. This is our key result.

2. *Benchmarking* against the density tree approach.

3. *Competitiveness*: Even if our technique produces significant improvements, it leaves open the question of these gains being competitive with task-specific techniques, e.g., cluster-explanation trees and prototype-based classifiers. These set of experiments affirmatively answer this question.

4. *Additional properties* - while these are not as rigorous as the previous groups, they highlight some interesting properties: (a) model size can be multivariate, and (b) it is possible to have different feature spaces between the oracle and the target models.

All experiments were performed on an Intel i7-7700HQ machine with 32 GB RAM.

Due to space constraints, only the key result - point 1 above - is presented in detail in the main paper (some aspects are relegated to the Appendix), while other findings are only summarized here, with details being provided in the Appendix.

### 4.1 EFFECTIVENESS OF OUR TECHNIQUE

We begin describing this set of experiments with the various settings.

#### 4.1.1 EXPERIMENT SETTINGS

We tested our technique across the following configurations:

1. Datasets: We use the following 13 publicly available standard classification datasets for our experiments: *cod-rna, ijcnn1, higgs, covtype.binary, phishing, a1a, pendigits, letter, Sensorless, senseit_aco, senseit_sei, covtype, connect-4*. These were obtained from the LIBSVM website (Chang & Lin, 2011a). For details, such as number of classes and extent of imbalance, please see §A.7.

   10000 instances from each dataset are used. The split ratio used in Algorithm 1 is $|D_{train}| : |D_{val}| : |D_{test}| :: 60 : 20 : 20$, where the splits are stratified wrt labels.

2. Interpretable model families: we use *Linear Probability Models (LPM)*[5] and the DTs (produced by the CART algorithm). These were picked as they are commonly considered interpretable Räz (2024).

   The notion of model size for LPMs is the number of non-zero coefficients, and sizes $\eta \in \{1, 2, ..., 15\}$ are explored (except for *cod-rna*, that has 8 features, and so we cannot have a sizes greater than 8).

   For DTs, the notion of size is depth. For a dataset, we first learn a tree (with no size constraints) with the highest *F1-macro* score using standard $5-$fold cross-validation. We refer to this as the optimal tree $T_{opt}$, and its depth as $depth(T_{opt})$. We then explore model sizes $\eta \in \{1, 2, ..., min(depth(T_{opt}), 15)\}$. Stopping early makes sense since the model is saturated in its learning at $depth(T_{opt})$; changing the input distribution is not helpful beyond this point.

3. Oracle families: As oracles we use *Random Forests (RF)* (Breiman, 2001) and *GBMs* (Friedman, 2001). They were learned using cross-validation or using a held-out set, to avoid overfitting, and were calibrated (Platt, 1999; Niculescu-Mizil & Caruana, 2005) for reliable probability estimates.

4. Optimization budget: For DTs, we use $T = 3000$, while for LPMs $T = 1000$ is used. These values were determined based on limited search. The budget for LPMs is lower since for multi-class datasets (7 of 13 here) we construct one-vs-all models which makes training LPMs time-consuming.

#### 4.1.2 METRICS

For various combinations of models and oracles, i.e., $\{LPM, DT\} \times \{GBM, RF\}$, we measure the percentage relative improvement in the *F1-macro* score (on the test set $D_{test}$) in terms of the baseline score $F1_{test}^{base}$ and the one produced by our model, $F1_{test}^*$:

$$\delta F1_{test} = \frac{100 \times (F1_{test}^* - F1_{test}^{base})}{F1_{test}^{base}} \tag{3}$$

We use the macro score since its not impacted by class imbalance.

In the interest of robustness we run **five trials per configuration**, i.e., a combination of dataset, oracle family, model family and size, and utilize the validation set to accept the model produced by our technique $M^*$. Specifically: indexing trials with $i$, we conduct an independent *t-test* on $\{F1_{val}^*\}_{1 \le i \le 5}$ and $\{F1_{val}^{base}\}_{1 \le i \le 5}$. The null hypothesis is that $M^*$ doesn't produce results different $M^{base}$. If we can reject the null at a significance of $p = 0.1$, we report $\delta F1_{test}$ as in Equation 3[6], else we report $\delta F1_{test} = 0$, i.e., we reject $M^*$. Here $\delta F1_{test} \in (-\infty, \infty)$; negative values are possible since we pick a model based on $D_{val}$ while we report based on $D_{test}$.

---

[5]We have not used the more common *Logistic Regression* because: (1) LPMs are considered more interpretable (Mood, 2010), and (2) LPMs results are indicative of behavior of linear models in general.

[6]The test scores from different trials are averaged first.

### 4.1.3 OBSERVATIONS

Table 1 shows a portion of the results in the interest of space - for complete results, and analysis of statistical significance (using the *Wilcoxon signed-rank* test (Wilcoxon, 1945)), please see §A.8.

Table 1: This table shows the average improvements, $\delta F1$, over **five runs** for the combinations **model={LPM, DT}** and **oracle=GBM**, for different model sizes. The improvements are measured relative to the model at the first iteration. Here, $\delta F1 \in (-\infty, \infty)$. Negative improvements are shown in underlined. **Complete results, including analysis of statistical significance, are presented in §A.8**.

| dataset | model_ora | size=1 | 2 | 3 | 4 | 5 | 6 | 7 | 8 | 9 | 10 | 11 | 12 | 13 | 14 | 15 |
|---|---|---|---|---|---|---|---|---|---|---|---|---|---|---|---|---|
| cod-rna | lpm_gbm | 1.39 | 12.53 | 14.76 | 15.73 | 14.97 | 12.00 | 0.00 | 0.08 | - | - | - | - | - | - | - |
|  | dt_gbm | 0.00 | 0.00 | 0.00 | 1.26 | 0.00 | 0.00 | 0.00 | 0.00 | -0.28 | 0.08 | - | - | - | - | - |
| ijcnn1 | lpm_gbm | -0.16 | 3.36 | 3.93 | 0.00 | 5.19 | 4.18 | 3.85 | 3.79 | 3.69 | 2.99 | 2.97 | 3.21 | 3.11 | 3.26 | 3.02 |
|  | dt_gbm | 1.96 | 12.00 | 10.15 | 11.37 | 10.63 | 7.18 | 3.63 | 4.52 | 2.91 | 1.78 | 1.93 | 2.29 | 1.47 | 2.26 | 0.00 |
| higgs | lpm_gbm | 29.29 | 17.80 | 11.40 | 6.56 | 3.06 | 2.68 | 3.16 | 2.90 | 2.67 | 2.82 | 2.65 | 1.79 | 2.62 | 2.19 | 1.63 |
|  | dt_gbm | 0.00 | 0.00 | 1.86 | 0.26 | 0.93 | 0.45 | - | - | - | - | - | - | - | - | - |
| covtype.binary | lpm_gbm | 76.52 | 66.39 | 29.17 | 12.51 | 9.18 | 5.28 | 4.94 | 4.56 | 3.92 | 3.56 | 3.62 | 3.31 | 2.59 | 2.83 | 2.39 |
|  | dt_gbm | 0.00 | 0.00 | 2.35 | 1.27 | 1.18 | 1.11 | 0.00 | 0.00 | 0.00 | - | - | - | - | - | - |
| phishing | lpm_gbm | 0.00 | 1.88 | 2.88 | 3.05 | 3.22 | 3.25 | 2.99 | 1.69 | 1.42 | 1.45 | 1.29 | 0.00 | 0.00 | 0.00 | 0.00 |
|  | dt_gbm | 0.00 | 0.00 | 0.00 | 0.07 | 0.39 | 0.00 | 0.28 | 0.22 | 0.44 | 0.23 | 0.00 | 0.00 | 0.00 | 0.00 | 0.00 |
| a1a | lpm_gbm | 0.00 | 2.55 | 7.58 | 8.98 | 8.40 | 8.03 | 8.90 | 8.23 | 8.17 | 7.90 | 5.96 | 7.10 | 6.97 | 6.18 | 5.73 |
|  | dt_gbm | 0.00 | 5.54 | 2.39 | 3.84 | 3.55 | 2.55 | 1.51 | 2.25 | 4.87 | - | - | - | - | - | - |
| pendigits | lpm_gbm | 51.39 | 23.44 | 16.18 | 8.95 | 8.84 | 6.63 | 4.86 | 1.83 | 2.27 | 2.16 | 2.44 | 2.16 | 3.33 | 2.97 | 2.73 |
|  | dt_gbm | 14.02 | 6.72 | 5.11 | 13.14 | 6.42 | 4.20 | 2.46 | 1.09 | 0.98 | 0.16 | -0.26 | 0.00 | 0.00 | 0.00 | 0.00 |
| letter | lpm_gbm | 57.06 | 48.48 | 59.85 | 29.76 | 36.09 | 19.27 | 20.37 | 16.08 | 17.55 | 15.16 | 17.26 | 16.51 | 18.46 | 17.19 | 15.55 |
|  | dt_gbm | 0.00 | 13.98 | 25.05 | 33.96 | 32.05 | 15.49 | 11.17 | 0.00 | 4.26 | 3.50 | 1.99 | 0.00 | 0.00 | 0.00 | 0.00 |
| Sensorless | lpm_gbm | 216.47 | 257.56 | 178.31 | 117.01 | 90.70 | 83.90 | 73.50 | 65.95 | 61.57 | 57.97 | 56.54 | 57.15 | 55.45 | 66.24 | 68.24 |
|  | dt_gbm | -0.01 | 42.42 | 68.13 | 44.38 | 17.39 | 10.32 | 1.82 | 1.44 | 0.79 | 0.64 | 0.41 | 0.12 | 0.00 | -0.02 | 0.34 |
| senseit_aco | lpm_gbm | 173.71 | 170.68 | 63.95 | 44.20 | 33.49 | 22.99 | 19.14 | 13.50 | 10.29 | 7.59 | 6.26 | 5.92 | 5.30 | 4.89 | 4.32 |
|  | dt_gbm | 14.89 | 0.00 | 3.71 | 2.32 | 4.85 | 0.81 | 0.00 | - | - | - | - | - | - | - | - |
| senseit_sei | lpm_gbm | 160.59 | 65.27 | 23.44 | 10.48 | 6.76 | 4.86 | 4.82 | 4.46 | 4.79 | 4.12 | 4.54 | 5.17 | 3.91 | 4.21 | 4.46 |
|  | dt_gbm | 2.66 | 1.01 | 3.49 | 2.29 | 0.95 | 1.30 | 1.37 | 0.00 | - | - | - | - | - | - | - |
| covtype | lpm_gbm | 36.87 | 49.24 | 12.78 | 11.21 | 7.84 | 7.15 | 7.15 | 8.07 | 7.70 | 8.25 | 10.94 | 8.35 | 4.37 | 8.77 | 5.84 |
|  | dt_gbm | 342.27 | 92.85 | 43.23 | 20.04 | 8.14 | 8.05 | 5.67 | 3.26 | 4.92 | 3.52 | 2.72 | 0.00 | 0.00 | 0.00 | 1.74 |
| connect-4 | lpm_gbm | 37.62 | 11.66 | 12.01 | 6.84 | 5.68 | 6.82 | 4.58 | 2.10 | 3.82 | 3.21 | 3.02 | 3.64 | 2.32 | 2.97 | 3.40 |
|  | dt_gbm | 89.33 | 29.23 | 20.20 | 12.10 | 9.73 | 9.88 | 7.82 | 7.43 | 0.57 | 4.61 | 1.08 | 3.35 | 2.23 | 1.15 | 1.55 |

We highlight some interesting trends:

1. The incidence of negative improvements is fairly low. Of course, this result set is incomplete, but referring to the complete set in §A.8, we note that only 13 of 690 non-null observations, or 1.88%, are negative. The average negative improvement is −0.24%.

2. As model size increases (left to right in Table 1), positive improvements (which can be high for small sizes, e.g., > 100%) tend to reduce. This makes intuitive sense since beyond a certain model size, when all informative patterns in the data have been captured, modifying the training distribution should not have much/any effect.

3. For DTs, the drop in improvements happen earlier than for LPMs. An intuitive explanation for this is that an unit increase in size for the LPM and DT do not lead to identical increase in capacity. DTs are non-linear models to begin with, and then, increasing their depth by one leads to a much larger increment in capacity, e.g., it doubles the number of leaves for a binary tree.

### 4.2 SUMMARY OF OTHER FINDINGS

In the interest of space, we summarize our other findings below:

1. Benchmarking against the density tree approach: we perform this comparison since it is the closest to ours in terms of methodology (see §2). The experiment settings are identical to the previous section, §4.1. While we present a detailed discussion in §A.9, the salient observations are:

   (a) We report the *scaled* difference in test *F1-macro* score improvement $(\delta F1^{ora} - \delta F1^{den}) / \max\{\delta F1^{den}, \delta F1^{ora}\}$. The denominator ensures a range of $[-1, 1]$, where a positive value is desiredAveraged over model sizes and datasets this value is 0.37 and 0.31 for LPMs and DTs respectively.

(b) An aggregate score like the above might be influenced by outlier improvement scores; so we also report the *percentage of times* we produce a better score. This is $81.38\%$ and $67.30\%$ for LPMs and DTs respectively.

For additional details, please see §A.9.

2. Competitiveness: We compare against techniques specialized for certain tasks, to see if our technique can elevate the performance of older techniques to be competitive:

   (a) On the task of *cluster explanation*, decision trees are constructed whose leaves represent clusters. Some specialized algorithms in the area are *ExShallow* Laber et al. (2021) and *Iterative Mistake Minimization (IMM)* Moshkovitz et al. (2020). While these are recent algorithms, we show that CART-based Breiman et al. (1984) DTs obtained by our technique *outperform* the more recent IMM.

   (b) We consider *prototype-based classification* where, in the interest of interpretability, we want a small number of prototypes. Here the notion of size is the number of prototypes. We show that using our technique improves the performance of a simple *Radial Basis Function Network (RBFN)* Broomhead & Lowe (1988) to perform similar to *Stochastic Neighbor Compression (SNC)* Kusner et al. (2014).

The *Mean Rank* is used as the primary metric, while the *Friedman* (Friedman, 1937) and *Wilcoxon signed-rank* tests are used to measure statistical significance. Please see §A.10 for details.

3. We also conducted these experiments to highlight some interesting properties:

   (a) It may be applied even in cases when model size is defined by more than one attribute, e.g., $max\_depth$ and $num\_trees$ in the case of GBMs. This is because Algorithm 1 delegates size enforcement to $train_{\mathcal{I},g}$. See §A.12 for details.

   (b) The technique works even if the oracle and the target model use different feature representations. This is because all that is required of the oracle are uncertainty scores, irrespective of how it arrives at them. We demonstrate this via a text classification task of predicting nationalities from surnames (Rao & McMahan, 2019). A *Gated Recurrent Unit (GRU)* (Cho et al., 2014) is used as the oracle. This is trained on a sequence of characters. The interpretable target model is a DT that uses character *n-grams* as input. See §A.13 for additional details.

# 5 LIMITATION: RUNTIMES

The cost of catering to non-differentiable loss functions, i.e., no gradient information, is high running times. Our experiments used *hyperopt* on account of its popularity and maturity, but this leads to high runtimes, e.g., for the *a1a* "dt_gbm" setting in Table 1, at our budget of $T = 3000$, the optimizer runs for close to an **hour**. But with a different surrogate model representation, e.g., *Gaussian Processes*, and with a judiciously picked acquisition function such as a noise-resilient version of *logEI* (Ament et al., 2023), the runtime can be reduced to $\sim$ **2 min**. These preliminary results are presented in §A.11. Our takeaway is that there exists a path to improving the runtime in future work.

# 6 CONCLUSION AND FUTURE WORK

In this work we presented a model-agnostic technique that obtains good size-accuracy trade-offs. This was empirically shown to perform well in diverse settings. Conveniently, there is only one hyperparameter to set (the number of iterations). Further, it can accommodate multivariate model sizes and can be used with differing feature spaces between the oracle and the interpretable models.

For future work, we think the following themes are meaningful: (a) Extension to differentiable models/losses for faster learning. Techniques such as *bilevel optimization*, e.g., Pedregosa (2016), might be useful here to learn instance weights directly, instead of a distribution. (b) We noted that improvements diminish with increasing model size (Section 4.1.2). It would be interesting to explore whether applying the technique separately to smaller models obtained from decomposing a larger model, e.g., subtrees within a DT, delays this effect. (c) Finally, exploring newer BayesOpt algorithms would be a good way to improve the running time for our algorithm - our current experiments (mentioned in §5) already indicate this to be a fruitful direction of study.

## ETHICS STATEMENT

We acknowledge that we have read, understood and adhere to the code of ethics provided at `https://iclr.cc/public/CodeOfEthics`. We declare that this paper faithfully represents research that was performed with rigor and integrity, and the claims presented here are substantiated by our experiments, which have been presented in detail in either the main paper or the Appendix.

We further declare that *Large Language Models (LLMs)* were not used in this work.

## REPRODUCIBILITY STATEMENT

We have uploaded our code as a supplementary material - this contains implementations of all ideas presented in paper.

The datasets used are publicly available and the versions available at the source mentioned in Section 4.1.1 were directly used without any modifications..

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

# A  APPENDIX

## A.1  1. UNCERTAINTY METRICS

Some other popular uncertainty metrics are:

1. **Least confident**: we calculate the extent of uncertainty w.r.t. the class we are most confident about:

$$u_M(x) = 1 - \max_{y_i \in \{1,2,...,C\}} M(y_i|x) \tag{4}$$

Here, we have $C$ classes, and $M(y_i|x)$ is the probability score produced by the model[7].

2. **Entropy**: this is the standard Shannon entropy measure calculated over class prediction confidences:

$$u_M(x) = \sum_{y_i \in \{1,2,...,C\}} -M(y_i|x) \log M(y_i|x) \tag{5}$$

We do not use the *least confident* metric since it completely ignores confidence distribution across labels. While *entropy* is quite popular, and does take into account the confidence distribution, we do not use it since it reaches its maximum for only points for which the classifier must be equally ambiguous about *all* labels; for datasets with many labels (one of our experiments uses a dataset with 26 labels - see Table 3) we may never reach this maximum.

Fig 3 visually shows what uncertainty values look like for the different metrics. Panel (a) displays a dataset with 4 labels. A probabilistic *linear Support Vector Machine (SVM)* is learned on this, and uncertainty scores corresponding to the metrics "margin", "least confident" and "entropy" are visualized in panels (b), (c) and (d) respectively. Darker shades of gray correspond to high uncertainty. Observe that only the "margin" metric in panel (b) achieves scores close to 1 at the two-label boundaries.

There is no best uncertainty metric in general, and the choice is usually application specific (Settles, 2009).

## A.2  SAMPLING FROM THE IBMM

Given our representation, the procedure to sample $N_s$ points, from a dataset $D$, using an oracle $M_O$ is shown in Algorithm 2. We also explain the steps below:

1. Determine partitioning over the $N_s$ points induced by the $DP$. We use the *Chinese Restaurant Process* Aldous (1985) for this. Let's assume this step produces $k$ partitions $\{c_1, c_2, ..., c_k\}$ and quantities $n_i \in \mathbb{N}$ where $\sum_{i=1}^{k} n_i = N$. Here, $n_i$ denotes the number of points that belong to partition $c_i$.

2. We determine the $Beta(A_i, B_i)$ component for each $c_i$ by sampling from the priors, i.e., $A_i \sim scale \times Beta(a, b)$ and $B_i \sim scale \times Beta(a', b')$.

3. Repeat for each $c_i$: for each instance-label pair $(x_j, y_j)$ in our training dataset, we calculate the oracle uncertainty score, $u_{M_O}(x_j)$. We then calculate $p_j = c \cdot Beta(u_{M_O}(x_j)|A_i, B_i)$. $c$ is a normalizing constant that scales the probabilities across instances to sum to 1. The quantities $p_j$ are used as sampling probabilities for various $(x_j, y_j)$, and $n_i$ points are sampled with replacement based on them.

## A.3  DEFAULT PARAMETERS

The optimizer we use, TPE, requires *box constraints*. Here we specify our search space for the optimization variables, $\Phi$ in Algorithm 1:

---

[7]The possibly confusing name "least confident" for this idea originated within the context of uncertainty sampling, where we are interested in sampling the most uncertain point, $x^* = \arg\min_x [\max_{y_i \in \{1,2,...,C\}} M(y_i|x)]$, which may be considered to be the instance with the "least most confident label".

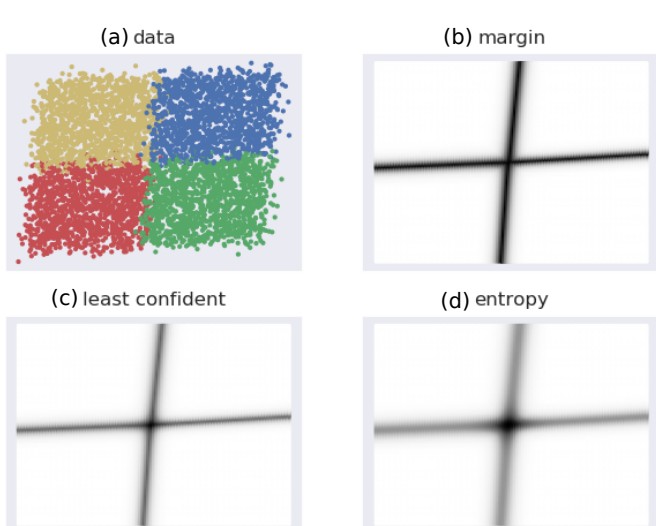

Figure 3: Visualizations of different uncertainty metrics. (a) shows a 4-label dataset on which linear SVM is learned. (b), (c), (d) visualize uncertainty scores based on different metrics, as per the linear SVM, where darker shades imply higher scores.

---

**Algorithm 2:** Sample based on uncertainties and $\Psi$

**Data:** Sample size $N_s$, oracle $M_O$, dataset $D = \{(x_i, y_i)\}_{i=1}^N$, IBMM parameters
    $\Psi = \{\alpha, a, b, a', b'\}$
**Result:** Sample $D'$, where $|D'| = N_s$

1   $D' = \{\}$ // assumed to be a multiset
2   $\{(c_1, n_1), (c_2, n_2), ..., (c_k, n_k)\} \leftarrow$ partition $N_s$ using the $DP$ // Here $\sum_{i=1}^k n_i = N_s$.
3   **for** $i \leftarrow 1$ **to** $k$ **do**
4      $A_i \sim scale \times Beta(a, b)$
5      $B_i \sim scale \times Beta(a', b')$
6      **for** $j \leftarrow 1$ **to** $N$ **do**
7          $p_j \leftarrow c \cdot Beta(u_{M_O}(x_j); A_i, B_i)$ // c is a normalizing constant s.t.
             $\sum_i^N c \cdot p_j = 1$.
8      **end**
9      $temp \leftarrow$ sample with replacement $n_i$ instance-label pairs based on $p_j$
10     $D' \leftarrow D' \uplus temp$ // $\uplus$ is the multiset sum
11 **end**
12 **return** $D'$

---

1. $p_o$: We want to allow the algorithm to pick an arbitrary fraction of samples from the original data; we set $p_o \in [0, 1]$.

2. $N_s$: We set $N_s \in [400, 10000]$. The lower bound ensures we have statistically significant results. The upper bound is set to a reasonably large value.

3. $\{a, b, a', b'\}$: Each of these parameters are allowed a range $[0.1, 10]$ to allow for a wide range of shapes for the component $Beta$ distributions.

4. $scale$: We fix $scale = 10000$ for our experiments, to allow for $A_i$ and $B_i$ to model skewed distributions where shape parameter large values might be required. For small values, the algorithm adapts by learning the appropriate $\{a, b, a', b'\}$.

5. $\alpha$: For a DP, $\alpha \in \mathbb{R}_{>0}$. We use a lower bound of $0.1$.

---

**Algorithm 3:** Learning interpretable model using oracle - reproduction of Algorithm 1.

---

**Data:** Dataset $D$, model size $\eta$, $train_{\mathcal{O},h}()$, $train_{\mathcal{I},g}()$, iterations $T$
**Result:** Optimal parameters $\Phi^*$, test set accuracy $s_{test}$ at $\Phi^*$, and interpretable model $M^*$ at $\Phi^*$

1 Create splits $D_{train}, D_{val}, D_{test}$ from $D$, stratified wrt labels. Here
  $|D_{train}| : |D_{val}| : |D_{test}| :: 60 : 20 : 20$.
2 $M_O \leftarrow train_{\mathcal{O},h}(D_{train}, *)$
3 **for** $t \leftarrow 1$ **to** $T$ **do**
4    $\Phi_t \leftarrow suggest(s_0, s_1, ...s_{t-1}, \Phi_0, \Phi_1, ..., \Phi_{t-1})$ `//` $s_0, \Phi_0$ `initialized at` $t=0$`,`
      `see text. Note:` $\Phi_t = \{\Psi_t, N_{s,t}, p_{o,t}\}$ `where` $\Psi_t = \{\alpha_t, a_t, b_t, a'_t, b'_t\}$`.`
5    $N_o \leftarrow p_{o,t} \times N_{s,t}$
6    $N_u \leftarrow N_{s,t} - N_o$
7    $D_o \leftarrow$ uniformly sample with replacement $N_o$ points from $D_{train}$
8    $D_u \leftarrow$ sample $N_u$ points from $D_{train}$ using the DP-based IBMM given current values for
      $N_u, M_O, D_{train}, \Psi_t$ `// see Algorithm A.2 for details`
9    $D_s \leftarrow D_o \uplus D_u$ `//` $D_o$`,` $D_u$ `are assumed to be multisets`
10    $M_t \leftarrow train_{\mathcal{I},g}(D_s, \eta)$
11    $s_t \leftarrow acc(M_t, D_{val})$
12 **end**
13 $t^* \leftarrow \arg\max_t \{s_1, s_2, ..., s_{T-1}, s_T\}$
14 $\Phi^* \leftarrow \Phi_{t^*}$
15 $M^* \leftarrow M_{t^*}$
16 $s_{test} \leftarrow acc(M^*, D_{test})$
17 **return** $\Phi^*, s_{test}, M^*$

---

To determine the upper bound, we rely on the following empirical relationship (Ohlssen et al., 2007) between the number of components $k$ and $\alpha$:

$$E[k|\alpha] \approx 5\alpha + 2 \tag{6}$$

We empirically estimated a fairly inclusive upper bound on the number of components to be 500, which provides us the $\alpha$ upper bound of 99.6. Thus, we use $\alpha \in [0.1, 99.6]$.

## A.4   NOTES ON THE MAIN ALGORITHM

We provide some additional details in reference to the main algorithm - Algorithm 1 - in the paper. For convenience, we reproduce the algorithm here, as Algorithm 3. Our notes follow:

1. We will consider the initialization to happen at $t = 0$, while the iterations range from 1 to $T$. $\Phi_0$ is set to: $\alpha = 0.1, a = 1, b = 1, a' = 1, b' = 1, N_s = |D_{train}|, p_o = 1$. A model is constructed based on $\Phi_0$ and a score $s_0$ is recorded. $(\Phi_0, s_0)$ serve as the history for the iteration at $t = 1$. The values for $\alpha, a, b, a', b'$ carry no significance and are arbitrary, since setting $p_o \to 1$ forces sampling only from the original distribution. Combined with $N_s = |D_{train}|$, this setting mimics the baseline, i.e., training the interpretable model without our algorithm, thus providing the optimizer with a good initial reference point in its search space.

2. The optimizer is represented by the function call $suggest()$ which takes as input all past parameter values and validation scores. $suggest()$ denotes a generic optimizer; not all optimizers require this extent of historical information.

3. While the training algorithm for the oracle, $train_{\mathcal{O},h}()$ is taken as input, a pre-constructed oracle $M_O$ may also be used. This would eliminate the oracle training step in line 2.

4. $acc()$ on the validation data, $D_{val}$, serves as both the objective and fitness function.

5. Evaluation on the test set, $D_{test}$ is done only once, in line 16, with the model that produces the best validation score.

6. Since we sample with replacement, both temporary datasets $D_o$ and $D_u$, procured from uniformly sampling the original training data and sampling based on uncertainties respectively, are multisets. Accordingly, line 9 uses the multiset sum operator $\uplus$ to combine them.

7. $M_t$ is created (line 10) with limited or no hyperparameter search using simple random validation, i.e., a stratified (by labels) random sample of size $0.2N_{s,t}$ is used as the validation set. A restricted search is performed because often hyperparameters are correlated with model size, and setting them to particular values would fail to produce a model of the required size $\eta$. As an example, consider DTs: setting a high threshold for the number of instances in a node for it be split (hyperparameter *min_samples_split* in *scikit-learn's* (Pedregosa et al., 2011) implementation) would produce only short trees.

We don't use cross-validation since at small values of $N_{s,t}$, the amount of training data, i.e., $(\frac{k-1}{k})N_{s,t}$ for $k$-folds, may become too small to obtain a good model. For example, for 3-folds, the training data size is $0.67N_{s,t}$. The data shortage problem can be addressed by increasing the number of folds, but that also increases the running time per iteration owing to the larger number of models that now need to be trained. As a practical compromise, we perform simple validation *thrice* and average the outcomes. This number is configurable, and may be decreased for models that are expensive to train.

8. Since the validation score $s_t$ (line 11) needs to be reliable, in our implementation we repeat lines 7-10 *thrice* and use the averaged validation score as $s_t$.

9. Class imbalance is accounted for in our implementation when training model $M_t$ in line 10. We either balance the data by sampling (this is the case with a *Linear Probability Model*), or an appropriate cost function is used to simulate balanced classes (this is the case with DTs and GBMs).

It is important to note here that $D_{val}$ *and* $D_{test}$ *are not modified by our algorithm in any way*, and therefore $s_t$ and $s_{test}$ measure the accuracy on the original distribution.

### A.5 SMOOTHING THE OPTIMIZATION LANDSCAPE

A practical consideration in our implementation is if we might facilitate finding the maxima $\Phi^*$ in Algorithm 1?

Since BayesOpt algorithms model the response surface of the actual objective function using a finite number of evaluations ($s_t$ in Algorithm 1), a certain degree of *smoothness* is assumed (Shahriari et al., 2016; Brochu et al., 2010). Here, the optimization variables $\Phi$ influence the objective value $s$ via this indirect chain: $\Phi_t \rightarrow D_s \rightarrow M_t \rightarrow s_t$ (symbols as in Algorithm 1), and for BayesOpt to work well, it is required that small changes in $\Phi_t$ result in small changes in $s_t$.

However, we have noticed that an oracle might produce uncertainty score distributions that are "spiky" or "jagged" - as an example, see the curve labelled "original" in Figure 4(a); which leads us to hypothesize that this principle is violated in general. A spiky distribution implies that small shifts $\Phi_t + \Delta\Phi_t$ may lead to sampling of instances with very different uncertainties; and since such instances may occur in regions far from those indicated by $\Phi_t$, they produce models with different class prediction behavior. This indirectly causes a disproportionate shift in $s_t$. While, in theory, a good BayesOpt algorithm should adapt to such problem characteristics, in practice they make the optimization problem harder, especially when the optimization budget is small.

To address this, we "flatten" the distribution[8] within $[0, 1]$. Our transformation is simple: we divide the interval $[0, 1]$ into $B$ bins, and map approximately $|D_{train}|/B$ uncertainty scores to each bin, while maintaining order between the original and mapped scores. Within a bin, the mapped scores are linearly spread across its range. This distributes the mapped scores approximately uniformly in the range $[0, 1]$. The algorithm is detailed in Algorithm 4.

Figure 4 visualizes the process of flattening. The original and modified uncertainty distributions for the datasets `Sensorless` and `covtype.binary` are shown in Figure 4(a) and 4(b) respectively.

While `Sensorless` appears to have a non-smooth distribution, and flattening here might help, this seems redundant for `covtype.binary`. *However, since this step is computationally cheap, we perform this for all our experiments, saving us the effort of assessing its need.*

---

[8]Distribution transformations have a long history in statistics, e.g., *power transforms* like the *Box-Cox* (Box & Cox, 1964) and *Yeo-Johnson* (Yeo & Johnson, 2000) transforms. Within ML, *Batch Normalization* (Ioffe & Szegedy, 2015) is a popular example of a distribution transformation applied to a loss landscape (Santurkar et al., 2018).

---

**Algorithm 4:** Flatten distribution of uncertainty scores $\{u(x_1), u(x_2), ..., u(x_N)\}$

---

**Data:** $\{u(x_1), u(x_2), ..., u(x_N)\}$, number of bins $B$
**Result:** $\{u'(x_1), u'(x_2), ..., u'(x_N)\}$

1   $bin\_size \leftarrow \lceil N/B \rceil, bin\_range \leftarrow 1/B$
2   $bin\_min \leftarrow [\,], bin\_max \leftarrow [\,]$
3   Let $sortedIndex(i) \in \{1, 2, ..., N\}$ be the index of $u(x_i)$ in the sequence of scores ordered by non-decreasing values.
4   **for** $j \leftarrow 1$ **to** $B$ **do**
5     $bin\_min[j] \leftarrow \min\{u(x_i)|i \in \{1, 2, ..., N\} \wedge sortedIndex(i) = j\}$
6     $bin\_max[j] \leftarrow \max\{u(x_i)|i \in \{1, 2, ..., N\} \wedge sortedIndex(i) = j\}$
7   **end**
8   **for** $i \leftarrow 1$ **to** $N$ **do**
9     $j \leftarrow sortedIndex(i)$
10    $bin\_num \leftarrow \lceil j/bin\_size \rceil$
11    $boundary\_low \leftarrow (bin\_num - 1) \times bin\_range + \delta$
12    $boundary\_high \leftarrow bin\_num \times bin\_range - \delta$
13    $u'(x_i) \leftarrow low + \frac{u(x_i) - bin\_min[j]}{bin\_max[j] - bin\_min[j]} \times (boundary\_high - boundary\_low)$
14   **end**
15   **return** $\{u'(x_1), u'(x_2), ..., u'(x_N)\}$

---

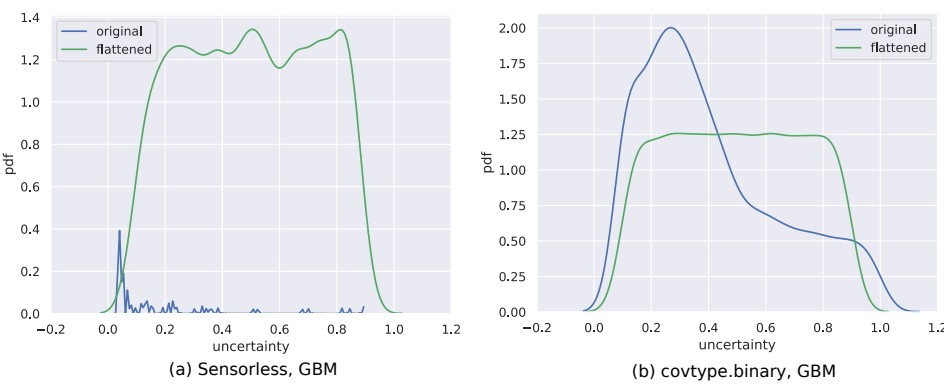

(a) Sensorless, GBM           (b) covtype.binary, GBM

Figure 4: Example of curve-flattening, for datasets (a) `Sensorless` and (b) `covtype.binary`. The uncertainty scores shown are obtained using the $GBM$ oracle.

Our transformation is invertible, which is useful in analyzing the observations from our experiments. Note however, it is not differentiable because of the discontinuities at the bin-boundaries; we also don't require this property.

The transformation affects line 7 in Algorithm 2. Instead of sampling based on the actual oracle uncertainty scores:

$$p_j \leftarrow Beta(u_{M_O}(x_j); A_i, B_i) \tag{7}$$

we sample based on the transformed uncertainty scores, $u'_{M_O}(x_j)$:

$$p_j \leftarrow Beta(u'_{M_O}(x_j); A_i, B_i) \tag{8}$$

In §A.6 we show that smoothing indeed has a positive effect.

A.6   EFFECT OF SMOOTHING

We first consider the question: does flattening (§A.5) help? Table 2 contrasts *improved* $F1$ scores obtained without (rows denoted as "original") and with (denoted "flattened") flattening the uncer-

Table 2: Improved scores averaged over three trials, shown for different parameter settings, with and without flattening. Here, Setting 1 is $\{max\_components = 500, scale = 10000\}$ and Setting 2 is $\{max\_components = 50, scale = 10\}$. "curr." signifies this is the current setting for our experiments in the main paper, while "low" signifies lower values of parameters. Highlighted cells indicate positive effect of flattening.

| dataset | dist. | Setting 1 (curr.) | | | Setting 2 (low) | | |
|---|---|---|---|---|---|---|---|
| | | 1 | 2 | 3 | 1 | 2 | 3 |
| Sensorless | original | 0.39 | 0.54 | 0.57 | 0.38 | 0.42 | 0.41 |
| | flattened | 0.44 | 0.53 | 0.55 | 0.43 | 0.54 | 0.59 |
| covtype.binary | original | 0.66 | 0.69 | 0.71 | 0.64 | 0.66 | 0.71 |
| | flattened | 0.68 | 0.73 | 0.73 | 0.65 | 0.71 | 0.71 |

tainty distribution. This is shown for the datasets `Sensorless` and `covtype.binary`, for model $size \in \{1, 2, 3\}$, with $model = LPM$ and $oracle = GBM$. Two different parameter settings are used: (a) In Setting 1, maximum allowed $Beta$ components are $500$ and $scale = 10000$ (b) Setting 2 looks at much lower values of these parameters where maximum allowed components is $50$ and $scale = 10$. The scores presented are the average over three trials.

We observe that while flattening influences results, other parameters determine the magnitude of its effect. At Setting 1, `Sensorless` is affected at $size = 1$ (flattening is better), but at higher sizes the differences seem to be from random variations across trials. At Setting 2 however, the differences are seen for $size \in \{1, 2, 3\}$ (flattening is better). For `covtype.binary` only $size = 2$ seems to be affected in either setting.

Recall we had noted in Figure 4 that the datasets `Sensorless` and `covtype.binary` have non-smooth and smooth uncertainty distributions respectively. The observations in Table 2 align well with the expectation that `Sensorless` is positively affected by the transformation, while results for `covtype.binary` remain mostly unchanged.

Based on these tests, we hypothesize that for non-smooth uncertainty distributions, flattening makes our technique robust across parameter settings. It does not affect smooth distributions in a significant way. Of course, rigorous and extensive tests are required to conclusively establish this effect.

## A.7 DATASETS

Table 3 provides details about the various datasets used in the experiments in §4. All of these are publicly available on the *LIBSVM* website (Chang & Lin, 2011a).

The "Label Entropy" column indicates how balanced a dataset is wrt its classes. For a dataset with $N$ instances and $C$ labels, this is calculated as:

$$\text{Label Entropy} = \sum_{j \in \{1, 2, ..., C\}} -p_j \log_C p_j \tag{9}$$

$$\text{Here, } p_j = \frac{|\{x_i | y_i = j\}|}{N}$$

Label Entropy $\in [0, 1]$, where values close to 1 denote the dataset is nearly balanced, and values close to 0 represent relative imbalance.

## A.8 VALIDATION RESULTS

An extended version of the results shown in Table 1 are presented here in Table 4. This shows results for all combinations of models and oracles: $\{LPM, DT\} \times \{GBM, RF\}$.

We also perform a *Wilcoxon signed-rank test* (Wilcoxon, 1945) to measure statistical significance. We use this test as it has been shown to be useful in comparing classifiers (Demšar, 2006; Benavoli et al., 2016; Japkowicz & Shah, 2011). Results are shown in figure 5 for the following test setup:

Table 3: We use the following datasets available on the LIBSVM website (Chang & Lin, 2011a). Their original source is mentioned in the "Description" column. 10000 instances from each dataset are used. A $train : val : test$ split ratio of $60 : 20 : 20$ is used for $D_{train}, D_{val}$ and $D_{test}$ in Algorithm 1. The splits are stratified wrt labels.

| S.No. | Dataset | Dimensions | # Classes | Label Entropy | Description |
|---|---|---|---|---|---|
| 1 | cod-rna | 8 | 2 | 0.92 | Predict presence of non-coding RNA common to a pair of RNA sequences, based on individual sequence properties and their similarity (Uzilov et al., 2006). |
| 2 | ijcnn1 | 22 | 2 | 0.46 | Time series data produced by an internal combustion engine is used to predict normal engine firings vs misfirings (Prokhorov, 2001). Transformations as in Chang & Lin (2001). |
| 3 | higgs | 28 | 2 | 1.00 | Predict if a particle collision produces Higgs bosons or not, based on collision properties (Baldi et al., 2014). |
| 4 | covtype.binary | 54 | 2 | 1.00 | Modification of the *covtype* dataset (see row 12), where classes are divided into two groups (Collobert et al., 2002). |
| 5 | phishing | 68 | 2 | 0.99 | Various website features are used to predict if the website is a *phishing* website (Mohammad et al., 2012). Transformations used as in Juan et al. (2016) |
| 6 | a1a | 123 | 2 | 0.80 | Predict whether a person makes over 50K a year, based on census data variables (Dua & Graff, 2017). Transformations as in Platt (1998). |
| 7 | pendigits | 16 | 10 | 1.00 | Classify handwritten digit samples into the digits 0-9 (Alimoglu & Alpaydin, 1996; Dua & Graff, 2017). |
| 8 | letter | 16 | 26 | 1.00 | Images of the capital letters A-Z were produced by random distortion of these characters from 20 fonts. The task is to classify these character images as one of the original letters (Michie et al., 1995). Transformations as in Hsu & Lin (2002). |
| 9 | Sensorless | 48 | 11 | 1.00 | Based on phase current measurements of an electric motor, predict different error conditions (Paschke et al., 2013). We use the transformations from Wang et al. (2018). |
| 10 | senseit_aco | 50 | 3 | 0.95 | Predict vehicle type using acoustic data gathered by a sensor network (Duarte & Hu, 2004). |
| 11 | senseit_sei | 50 | 3 | 0.94 | Predict vehicle type using seismic data gathered by a sensor network (Duarte & Hu, 2004). |
| 12 | covtype | 54 | 7 | 0.62 | Predict forest cover type from cartographic variables (Dean & Blackard, 1998; Dua & Graff, 2017). |
| 13 | connect-4 | 126 | 3 | 0.77 | Predict if the first player wins, loses or draws, based on board positions of the board game *Connect Four* (Dua & Graff, 2017). |

Table 4: This table shows the average improvements, $\delta F1$, over five runs for different combinations of models and oracles: $\{LPM, DT\} \times \{GBM, RF\}$. This is an extended version of the results in Table 1. The improvements are measured relative to the model at the first iteration. The best improvement for a model size and oracle is indicated in bold. Here, $\delta F1 \in (-\infty, \infty)$. Negative improvements are shown in underlined.

| dataset | model_ora | 1 | 2 | 3 | 4 | 5 | 6 | 7 | 8 | 9 | 10 | 11 | 12 | 13 | 14 | 15 |
|---|---|---|---|---|---|---|---|---|---|---|---|---|---|---|---|---|
| cod-rna | lpm_gbm | 1.39 | 12.53 | **14.76** | **15.73** | 14.97 | 12.00 | 0.00 | **0.08** | - | - | - | - | - | - | - |
| | lpm_rf | 2.66 | **13.91** | 14.69 | 15.34 | **16.06** | **12.49** | **8.30** | 0.00 | - | - | - | - | - | - | - |
| | dt_gbm | **0.00** | **0.00** | 0.00 | 1.26 | 0.00 | **0.00** | 0.00 | 0.00 | _-0.28_ | 0.08 | - | - | - | - | - |
| | dt_rf | 0.00 | 0.00 | **1.78** | **2.28** | **0.39** | _-0.02_ | **0.17** | **0.47** | **0.00** | **0.72** | - | - | - | - | - |
| ijcnn1 | lpm_gbm | _-0.16_ | **3.36** | **3.93** | 0.00 | **5.19** | **4.18** | **3.85** | **3.79** | **3.69** | 2.99 | **2.97** | **3.21** | 3.11 | 3.26 | 3.02 |
| | lpm_rf | **0.19** | 2.80 | 3.36 | **3.65** | 3.33 | 1.94 | 3.58 | 3.30 | 3.46 | **3.81** | 2.66 | **4.65** | **3.99** | **3.82** | **4.85** |
| | dt_gbm | 1.96 | 12.00 | **10.15** | **11.37** | **10.63** | 7.18 | 3.63 | **4.52** | **2.91** | **1.78** | **1.93** | **2.29** | 1.47 | **2.26** | 0.00 |
| | dt_rf | **4.06** | **12.10** | 8.95 | 10.75 | 10.13 | **8.25** | **5.38** | 2.46 | 2.63 | 1.25 | 1.46 | 1.37 | **1.91** | 0.00 | **1.38** |
| higgs | lpm_gbm | **29.29** | **17.80** | 11.40 | 6.56 | 3.06 | 2.68 | **3.16** | **2.90** | **2.67** | **2.82** | 2.65 | 1.79 | 2.62 | 2.19 | 1.63 |
| | lpm_rf | 26.71 | 17.29 | **15.06** | **10.60** | **5.35** | **4.04** | 2.35 | 2.03 | 1.66 | 1.89 | **2.91** | **2.94** | **3.31** | **2.58** | **2.22** |
| | dt_gbm | 0.00 | 0.00 | **1.86** | 0.26 | 0.93 | 0.45 | - | - | - | - | - | - | - | - | - |
| | dt_rf | **4.04** | **1.26** | 1.74 | **1.32** | **1.54** | **0.91** | - | - | - | - | - | - | - | - | - |
| covtype.binary | lpm_gbm | 76.52 | **66.39** | **29.17** | **12.51** | **9.18** | **5.28** | **4.94** | **4.56** | **3.92** | **3.56** | **3.62** | **3.31** | **2.59** | **2.83** | **2.39** |
| | lpm_rf | **96.77** | 63.38 | 14.36 | 9.61 | 6.79 | 3.94 | 2.93 | 2.81 | 2.96 | 2.84 | 2.31 | 2.26 | 2.00 | 2.43 | 2.22 |
| | dt_gbm | **0.00** | **0.00** | **2.35** | 1.27 | 1.18 | 1.11 | 0.00 | 0.00 | 0.00 | - | - | - | - | - | - |
| | dt_rf | 0.00 | 0.00 | 2.10 | **2.33** | **2.44** | **2.39** | **1.84** | **2.19** | **1.65** | 0.70 | - | 0.89 | - | - | - |
| phishing | lpm_gbm | **0.00** | 1.88 | 2.88 | 3.05 | 3.22 | 3.25 | 2.99 | 1.69 | 1.42 | **1.45** | **1.29** | 0.00 | 0.00 | 0.00 | 0.00 |
| | lpm_rf | 0.00 | **2.14** | **3.29** | **3.22** | **3.59** | **3.79** | **3.29** | **2.05** | **1.42** | 1.44 | 1.24 | **1.23** | **1.16** | **1.26** | **1.02** |
| | dt_gbm | **0.00** | 0.00 | **0.00** | 0.07 | 0.39 | 0.00 | 0.28 | 0.22 | **0.44** | **0.23** | 0.00 | **0.00** | **0.00** | 0.00 | **0.00** |
| | dt_rf | 0.00 | **0.72** | 0.00 | **0.57** | 0.00 | _-0.17_ | 0.13 | **0.48** | 0.13 | 0.05 | **0.03** | _-0.03_ | _-0.28_ | 0.00 | _-0.16_ |
| a1a | lpm_gbm | **0.00** | 2.55 | 7.58 | 8.98 | 8.40 | 8.03 | 8.90 | 8.23 | 8.17 | 7.90 | 5.96 | 7.10 | 6.97 | 6.18 | 5.73 |
| | lpm_rf | 0.00 | **4.17** | **8.81** | **9.92** | **9.88** | **9.47** | **8.99** | **9.31** | **9.19** | **9.26** | **9.33** | **8.25** | **7.15** | **7.55** | **7.98** |
| | dt_gbm | **0.00** | 5.54 | 2.39 | 3.84 | **3.55** | 2.55 | 1.51 | 2.25 | 4.87 | - | - | - | - | - | - |
| | dt_rf | 0.00 | **6.44** | **3.36** | **5.60** | 3.40 | **5.94** | **6.06** | **4.97** | **4.89** | 4.01 | 4.73 | 5.21 | - | - | 4.53 |
| pendigits | lpm_gbm | **51.39** | **23.44** | 16.18 | **8.95** | **8.84** | **6.63** | 4.86 | 1.83 | 2.27 | 2.16 | **2.44** | 2.16 | **3.33** | 2.97 | 2.73 |
| | lpm_rf | 46.28 | 22.74 | **21.72** | 8.80 | 8.47 | 6.29 | **6.48** | 1.69 | **3.03** | **2.79** | 2.34 | **2.68** | 2.70 | **3.02** | 0.00 |
| | dt_gbm | 14.02 | **6.72** | 5.11 | 13.14 | 6.42 | 4.20 | 2.46 | **1.09** | 0.98 | **0.16** | _-0.26_ | **0.00** | **0.00** | **0.00** | **0.00** |
| | dt_rf | **21.46** | 4.18 | **5.22** | **14.51** | **7.36** | **4.55** | **2.86** | 0.00 | 0.00 | 0.00 | **0.00** | 0.00 | 0.00 | 0.00 | 0.00 |
| letter | lpm_gbm | 57.06 | 48.48 | 59.85 | **29.76** | **36.09** | 19.27 | 20.37 | 16.08 | 17.55 | 15.16 | 17.26 | 16.51 | 18.46 | 17.19 | 15.55 |
| | lpm_rf | **61.06** | **65.34** | **64.26** | 23.69 | 35.20 | **26.15** | **22.10** | **20.74** | **20.91** | **20.31** | **19.28** | **21.40** | **20.77** | **19.39** | **18.18** |
| | dt_gbm | 0.00 | **13.98** | 25.05 | **33.96** | 32.05 | 15.49 | **11.17** | 0.00 | **4.26** | **3.50** | **1.99** | 0.00 | 0.00 | 0.00 | **0.00** |
| | dt_rf | 0.00 | 12.21 | **28.67** | 33.47 | **33.51** | **18.41** | 8.10 | 0.00 | 1.84 | 1.21 | 1.31 | **0.67** | **0.61** | **0.11** | _-0.08_ |
| Sensorless | lpm_gbm | 216.47 | **257.56** | **178.31** | **117.01** | **90.70** | **83.90** | **73.50** | **65.95** | 61.57 | 57.97 | 56.54 | 57.15 | 55.45 | 66.24 | 68.24 |
| | lpm_rf | **224.18** | 210.28 | 134.44 | 115.00 | 85.85 | 74.96 | 66.77 | 61.10 | **66.88** | **64.65** | **69.00** | **70.09** | **72.91** | **80.14** | **82.15** |
| | dt_gbm | _-0.01_ | 42.42 | **68.13** | 44.38 | 17.39 | 10.32 | 1.82 | **1.44** | **0.79** | 0.64 | 0.41 | 0.12 | **0.00** | _-0.02_ | **0.34** |
| | dt_rf | **0.00** | **52.54** | 57.10 | **44.61** | 16.63 | 6.19 | **2.19** | 0.96 | 0.51 | 0.00 | **0.48** | **0.33** | 0.00 | **0.00** | 0.10 |
| senseit_aco | lpm_gbm | 173.71 | 170.68 | 63.95 | **44.20** | 33.49 | 22.99 | 19.14 | 13.50 | 10.29 | 7.59 | 6.26 | 5.92 | **5.30** | 4.89 | 4.32 |
| | lpm_rf | **177.67** | **181.26** | **79.86** | 42.86 | **37.60** | **28.80** | **23.75** | **19.06** | **13.91** | **10.74** | **8.48** | **6.09** | 5.20 | **5.32** | **4.62** |
| | dt_gbm | 14.89 | 0.00 | **3.71** | 2.32 | **4.85** | 0.81 | **0.00** | - | - | - | - | - | - | - | - |
| | dt_rf | **20.03** | **2.54** | 3.64 | **5.91** | 3.34 | **2.63** | 0.00 | - | - | - | - | - | - | - | - |
| senseit_sei | lpm_gbm | 160.59 | **65.27** | 23.44 | 10.48 | 6.76 | 4.86 | 4.82 | 4.46 | 4.79 | 4.12 | 4.54 | **5.17** | 3.91 | 4.21 | **4.46** |
| | lpm_rf | **165.98** | 63.72 | **31.58** | **14.94** | **9.07** | **5.79** | **4.95** | **5.07** | **5.24** | **4.70** | **4.60** | 3.74 | **4.30** | **4.35** | 4.35 |
| | dt_gbm | **2.66** | **1.01** | **3.49** | **2.29** | **0.95** | **1.30** | **1.37** | 0.00 | - | - | - | - | - | - | - |
| | dt_rf | 2.33 | 0.00 | 3.36 | 1.65 | 0.87 | 0.00 | _-1.23_ | - | - | - | - | - | - | - | - |
| covtype | lpm_gbm | 36.87 | **49.24** | **12.78** | **11.21** | 7.84 | 7.15 | 7.15 | 8.07 | 7.70 | 8.25 | **10.94** | **8.35** | 4.37 | 8.77 | 5.84 |
| | lpm_rf | 32.15 | 39.49 | 10.49 | 8.53 | **8.11** | **8.59** | **9.61** | **11.99** | **11.22** | **9.91** | 8.47 | 8.16 | **10.34** | **13.76** | **12.92** |
| | dt_gbm | 342.27 | 92.85 | 43.23 | **20.04** | 8.14 | **8.05** | **5.67** | 3.26 | **4.92** | **3.52** | **2.72** | 0.00 | 0.00 | 0.00 | **1.74** |
| | dt_rf | **354.45** | **98.94** | **50.87** | 14.10 | **9.46** | 7.38 | 4.76 | **4.20** | 0.94 | 1.81 | 2.30 | **0.71** | _-0.37_ | 0.00 | 0.00 |
| connect-4 | lpm_gbm | **37.62** | 11.66 | 12.01 | 6.84 | 5.68 | 6.82 | 4.58 | 2.10 | 3.82 | 3.21 | **3.02** | **3.64** | **2.32** | **2.97** | **3.40** |
| | lpm_rf | 33.77 | **12.99** | **17.60** | **14.66** | **15.91** | **10.73** | **6.38** | **5.35** | **7.07** | **6.98** | 2.84 | 3.14 | 2.09 | 2.52 | 2.46 |
| | dt_gbm | 89.33 | **29.23** | 20.20 | **12.10** | 9.73 | 9.88 | 7.82 | 7.43 | 0.57 | 4.61 | 1.08 | **3.35** | 2.23 | **1.15** | **1.55** |
| | dt_rf | **113.71** | 21.91 | **20.52** | 11.23 | **16.86** | **10.96** | **10.64** | **9.11** | **6.51** | **5.88** | **6.76** | 2.16 | **2.97** | 0.61 | 0.00 |

1. We divide the analysis by model size. This is because size strongly influences $\delta F1$ (as in Table 4).

2. Normalized model sizes are used. Binning of model sizes is done using *Sturges rule* (Sturges, 1926).

3. The *one-sided* version of the *paired* test is performed for each bin, where pairs of scores $F1^{base}$ and $F1^*$ for a dataset, for models with sizes assigned to the bin, are compared. In cases were where multiple model sizes for a dataset fall within the same bin, $F1^{base}$ and $F1^*$ are first averaged and then compared.

4. The following hypotheses are tested:

   - **H$_0$**, null hypothesis: accuracies of models produced by our technique are not better.
   - **H$_1$**, alternate hypothesis: accuracies of models trained using the oracle are better.

   *p-values* are shown for each bin. Small *p-values* favor **H$_1$**, i.e., our algorithm.

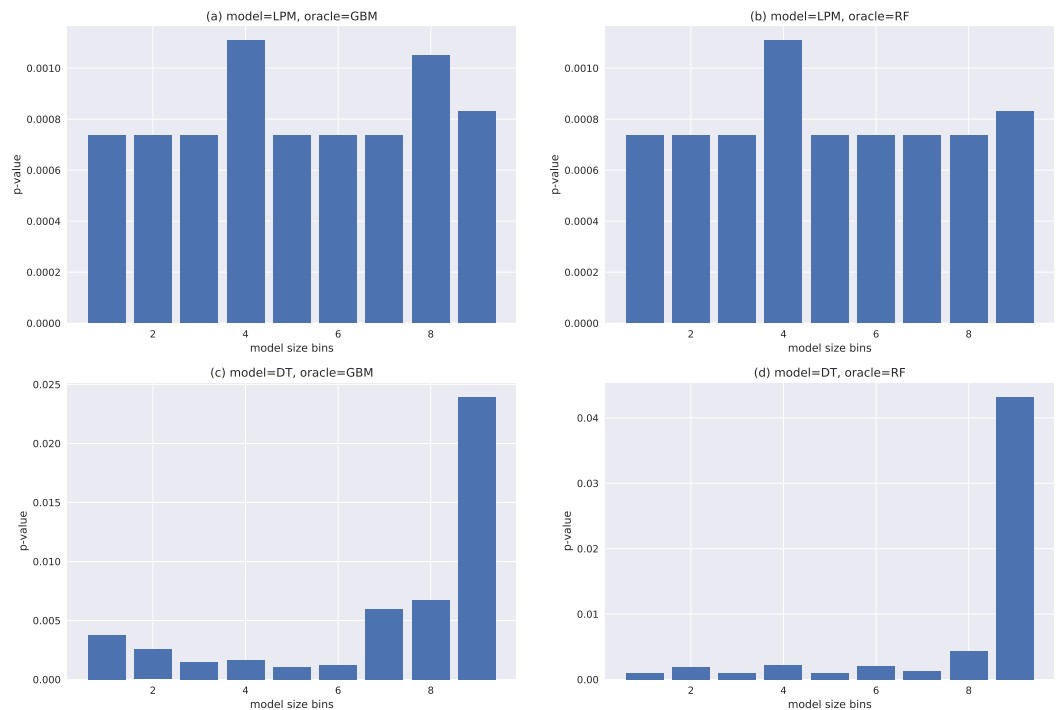

Figure 5: These plots show the *p-values* for the Wilcoxon signed-rank test, with the null hypothesis $H_0$: using the oracle does *not* produce better F1 test scores. The bin boundaries are selected using the *Sturges* rule (Sturges, 1926). Low *p-values* favor our algorithm.

5. Scores of $\delta F1 = 0$ are split equally between positive and negative ranks[9].

### A.9 COMPARISON WITH THE APPROACH BASED ON DENSITY TREES

As mentioned in §4.2 we benchmark against the density tree technique Ghose & Ravindran (2020) because that's the closest in terms of methodology. Their metric is slightly different from ours. Instead of reporting results for $F1^*$, they report them for $max(F1^*, F1^{base})$. This is an "outcome-centric" view[10], where you can't do worse than your best model. For this case, $\delta F1_{test} \in [0, \infty)$. We also follow this scoring scheme in this section to match their reporting.

We report two scores for comparison (*den* and *ora* denote density trees and our oracle based technique respectively):

1. To compare improvements, we use the *Scaled Difference in Improvement (SDI)*:

$$SDI = \begin{cases} (\delta F1^{ora} - \delta F1^{den})/H, & \text{if } H > 0 \\ 0, & \text{if } H = 0 \end{cases} \tag{10}$$

$$\text{where } H = \max\left\{\delta F1^{den}, \delta F1^{ora}\right\}$$

Here $\delta F1^{ora}$ and $\delta F1^{den}$ are the improvements from our technique and by using density trees, respectively. The scaling wrt $H$ ensures that $SDI \in [-1, 1]$ making it convenient to interpret. Note that $H \geq 0$ since both $\delta F1^{ora} \geq 0$ and $\delta F1^{den} \geq 0$ in the current scoring

---

[9]The `zplit` option in `https://numpy.org/doc/stable/reference/generated/numpy.histogram_bin_edges.html` is used.

[10]Another reason provided is that with a sufficient budget the optimizer will eventually learn to set $p_o = 1$, thus emulating $M^{base}$ exactly, if $M^{base}$ is indeed the best possible model. In this case $\delta F1 = 0$ as per Equation 3.

scheme. For brevity, we average the $SDI$ scores at the level of a dataset, across model sizes, for a given model and oracle. This averaged score is denoted by $\overline{SDI}$, and this is what we report.

2. Since $\overline{SDI}$ is aggregated over model sizes, we also report the percentage of times $\delta F1^{ora} > \delta F1^{den}$ across these model sizes. This is denoted as $pct\_better$

All $\delta F1^{ora}$ and $\delta F1^{den}$ scores used are the *averaged over five runs*.

We consider our approach to be better if $\overline{SDI} > 0$ *and* $pct\_better > 50\%$. These scores are shown in Table 5. Since the density trees approach lacks a notion of an oracle, we present results for GBMs and RFs separately. Numbers that represent superior performance by density trees are underlined. Note also the two special groupings:

- **ANY**: For each model size, the $SDI$ score considered is the higher of the ones obtained from using the $GBM$ or $RF$ as oracles. The $\overline{SDI}$ and $pct\_better$ scores are computed based on these scores. This grouping represents the ideal way to use our technique in practice: try multiple oracles and pick the best.
- **OVERALL**: This averages results across datasets, to provide an aggregated view.

The cells identified by **OVERALL** *and* **ANY** provide comparison numbers aggregated over datasets, model sizes and oracles.

Table 5: LPM, DT compared to the Density Tree approach. All $\delta F1^{ora}$ and $\delta F1^{den}$ scores used are the *average over five runs*. Cases where density trees fare better are underlined. The line in the middle separates binary class datasets (top) from multi-class ones (bottom).

| dataset | LPM | | | DT | | |
|---|---|---|---|---|---|---|
| | GBM | RF | **ANY** | GBM | RF | **ANY** |
| cod-rna | -0.38, 0.00% | -0.45, 0.00% | -0.33, 0.00% | 0.51, 60.00% | 0.50, 70.00% | 0.65, 80.00% |
| ijcnn1 | 0.06, 66.67% | 0.11, 80.00% | 0.20, 93.33% | 0.23, 53.33% | 0.68, 100.00% | 0.68, 100.00% |
| higgs | -0.07, 40.00% | -0.07, 40.00% | 0.04, 46.67% | 0.23, 50.00% | 0.61, 83.33% | 0.61, 83.33% |
| covtype.binary | -0.16, 40.00% | -0.33, 13.33% | -0.15, 40.00% | 0.23, 66.67% | 0.26, 72.73% | 0.38, 81.82% |
| phishing | 0.30, 80.00% | 0.37, 86.67% | 0.38, 86.67% | 0.11, 26.67% | -0.00, 26.67% | 0.23, 46.67% |
| a1a | -0.03, 60.00% | 0.13, 66.67% | 0.13, 66.67% | -0.06, 44.44% | 0.43, 75.00% | 0.52, 83.33% |
| pendigits | 0.59, 100.00% | 0.59, 93.33% | 0.62, 100.00% | 0.23, 60.00% | 0.16, 46.67% | 0.25, 60.00% |
| letter | 0.79, 100.00% | 0.81, 100.00% | 0.81, 100.00% | 0.02, 33.33% | -0.34, 13.33% | 0.06, 40.00% |
| Sensorless | 0.64, 100.00% | 0.65, 100.00% | 0.66, 100.00% | -0.23, 20.00% | -0.39, 20.00% | -0.23, 20.00% |
| senseit_aco | 0.55, 100.00% | 0.63, 100.00% | 0.63, 100.00% | 0.50, 85.71% | 0.37, 75.00% | 0.39, 75.00% |
| senseit_sei | 0.61, 100.00% | 0.66, 100.00% | 0.67, 100.00% | -0.25, 42.86% | 0.51, 100.00% | 0.51, 100.00% |
| covtype | 0.20, 80.00% | 0.39, 93.33% | 0.43, 100.00% | 0.26, 66.67% | 0.16, 66.67% | 0.40, 80.00% |
| connect-4 | 0.23, 73.33% | 0.24, 66.67% | 0.38, 86.67% | -0.23, 33.33% | -0.13, 53.33% | 0.08, 66.67% |
| **OVERALL** | 0.28, 75.00% | 0.32, 75.00% | 0.37, 81.38% | 0.10, 47.06% | 0.16, 57.23% | 0.31, 67.30% |

The predominance of non-underlined values indicate that our technique performs better in most settings. In both cases, the **OVERALL** +**ANY** entries indicate that our technique works better on average - in terms of both the extent of improvement $\overline{SDI}$ and its frequency $pct\_better$.

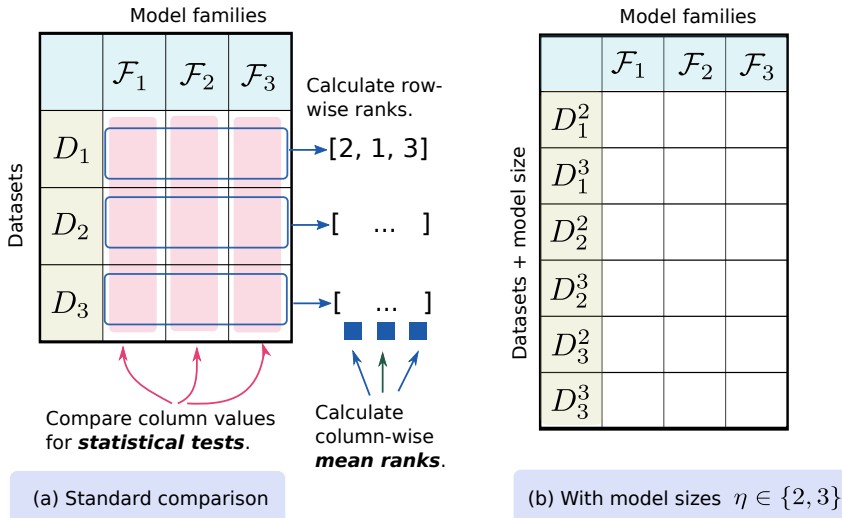

Figure 6: (a) shows a standard measurement scheme with datasets in rows and model families in columns. Statistical tests are performed on the column values. Row-wise ranks are first computed for calculating the mean rank. (b) To account for model sizes, we allow rows to represent combinations of datasets and model sizes. See text for details.

### A.10 EVALUATING COMPETITIVENESS

As mentioned in the main paper in §4.2, we consider the following tasks for evaluating competitiveness:

1. Building cluster explanation trees.
2. Prototype-based classification.

For evaluation on each of these tasks, we follow a common theme: (a) first, we show that a traditional technique is almost always not as good as newer and specialized techniques, and, (b) then we show that its performance may be radically improved by learning the training distribution. Collectively, these evaluations show that the strategy of learning the training distribution is both *general* - may be applied to different tasks, models, notions of model sizes - and *effective* - results in competitive performance. We first detail our measurement strategy.

#### A.10.1 MEASUREMENT

While each task-specific section contains a detailed discussion on the experiment setup, we discuss some common aspects here:

1. To compare model families $\mathcal{F}_1, \mathcal{F}_2, \mathcal{F}_3$, each of which is, say, used to construct models for different sizes $\eta \in \{2, 3\}$, for datasets $D_1, D_2, D_3$, we use the *mean rank*, and support our conclusions with statistical tests such as the *Friedman* (Friedman, 1937) and *Wilcoxon signed-rank* (Wilcoxon, 1945) tests[11].

   Typically mean rank is used to compare model families based on their accuracies across datasets - which, ignoring model sizes, may be visualized as a $3 \times 3$ table here, with rows representing datasets, and columns denoting model families - see Figure 6(a). An entry such as "$D_2, \mathcal{F}_3$" represents the accuracy (or some other metric) of a model from family $\mathcal{F}_3$ on dataset $D_2$. Models are ranked on a per-dataset basis, i.e., row-wise, and the average ranks (computed per family, i.e., column-wise) are reported (lower is better). For statistical tests, the column values are directly used.

---

[11]The *Wilcoxon signed-rank* test was used here since it has been advocated by various studies for measuring classification performance (Demšar, 2006; Benavoli et al., 2016; Japkowicz & Shah, 2011).

However, we have an additional factor here - the model size. To avoid inventing a custom metric, we assimilate it in the previous scheme by using the combination of datasets *and* model sizes as a row - see Figure 6(b). We think of such combinations as "pseudo-dataset" entries, i.e., now we have a $6 \times 3$ table, with rows for $D_1^2, D_1^3, D_2^2, D_2^3, D_3^2, D_3^3$, and same columns as before. The entry for "$D_1^2, \mathcal{F}_3$" indicates the accuracy of a model of size 2 from family $\mathcal{F}_3$ on dataset $D_1$.

Effectively, now the comparisons automatically account for model size since we use pseudo-datasets instead of datasets.**Note** that no new datasets are being created - we are merely defining a convention to include model size in the familiar dataset-model cross-product table.

2. For *each* model family, model size and dataset combination (essentially a cell in this cross-product table), models are constructed multiple times (we refer to these as multiple "trials"), and their scores are averaged. Five trials were used in our experiments.

### A.10.2   EXPLAINABLE CLUSTERING

The first task we investigate is the problem of *Explainable Clustering*. Introduced by Moshkovitz et al. (2020), the goal is to explain cluster allocations as discovered by techniques such *k-means* or *k-medians*. This is achieved by constructing axis-aligned decision trees with leaves that either exactly correspond to clusters, e.g., *Iterative Mistake Minimization (IMM)* Moshkovitz et al. (2020), or are proper subsets, e.g., *Expanding Explainable k-Means Clustering (ExKMC)* Frost et al. (2020). We consider the former case here, i.e., a tree must possess exactly $k$ leaves to explain $k$ clusters.

For a specific clustering $C$, let $C(x_i)$ denote the assigned cluster for an instance $x_i, i = 1...N$, where $C(x_i) \in \{1, 2, ..., k\}$, and the cluster centroids by $\mu_j, j = 1, ..., k$. The cost of clustering $J$ is then given by:

$$J = \frac{1}{N} \sum_{j=1}^{k} \sum_{\{x_i | C(x_i) = j\}} ||x_i - \mu_j||_2^2 \tag{11}$$

In the case of an explanation trees with $k$ leaves, $\mu_j$ are centroids of leaves. Cluster explanation techniques attempt to minimize this cost.

The price of explainability maybe measured as the *cost ratio*[12]:

$$\text{cost ratio} = \frac{J_{Ex}}{J_{KM}} \tag{12}$$

Here $J_{Ex}$ is the cost achieved by an explanation tree, and $J_{KM}$ is the cost obtained by a standard k-means algorithm. It assumes values in the range $[1, \infty]$, where the lowest cost is obtained when using k-means, i.e., $J_{Ex}$ and $J_{KM}$ are the same.

One may also indirectly minimize the cost in the following manner: use k-means to produce a clustering, use the cluster allocations of instances as their labels, and then learn a standard decision tree for classification, e.g., CART. This approach has been shown to be often outperformed by tree construction algorithms that directly minimize the cost in Equation 11, e.g., IMM.

⬦ **Algorithms and Hyperparameters**
 The algorithms we compare and their hyperparameter settings are as follows:

1. **Iterative Mistake Minimization (IMM)** Moshkovitz et al. (2020): This generates a decision tree via greedy partitioning using a criterion that minimizes number of mistakes at each split (the number of points separated from their corresponding reference cluster center). There are no parameters to tune. We used the implementation available here: https://github.com/navefr/ExKMC, which internally uses the reference implementation for IMM.

2. **ExShallow** Laber et al. (2021): Here, the decision tree construction explicitly accounts for minimizing explanation complexity while targeting a low cost ratio. The trade-off between

---

[12]This is referred to as the *cost ratio* in Frost et al. (2020), *price of explainability* in Moshkovitz et al. (2020) and *competitive ratio* in Makarychev & Shan (2022).

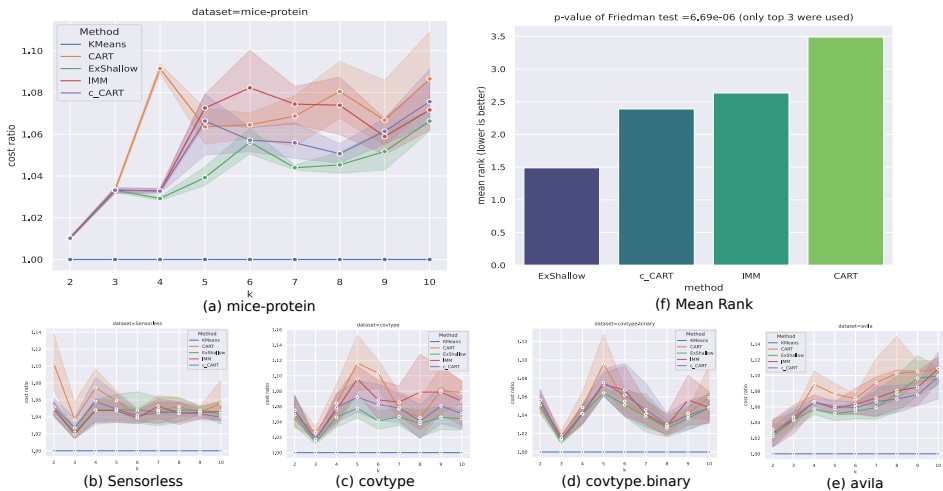

Figure 7: Comparisons over explainable clustering algorithms are shown. (a) shows the comparison for a specific dataset *mice-protein*. (b), (c), (d) and (e) show comparisons over other datasets - miniaturized to fit the page. (f) shows mean ranks of these techniques over five datasets across model sizes; the Friedman test is conducted over the **top three** techniques only, with $p = 6.688 \times 10^{-6}$.
.

    clustering cost and explanation size is controlled via a parameter $\lambda$. This is set as $\lambda = 0.03$ in our experiments; this value is used in the original paper for various experiments. We used the reference implementation available here: `https://github.com/lmurtinho/ShallowTree`.

3. **CART**: We use CART Breiman et al. (1984) as the traditional model to compare, and maximize the classification accuracy for predicting clusters, as measured by the F1-macro score. The implementation in *scikit* Pedregosa et al. (2011) is used. During training, we set the following parameters: (a) the maximum number of leaves (this represents *model size $\eta$* here) is set to the number of clusters $k$, and (b) the parameter *class_weight* is set to *"balanced"* for robustness to disparate cluster sizes. Results for CART are denoted with label **CART**. We then apply our technique to CART; these results are denoted as **c_CART**. We set $T = 2000$, and use default settings for other parameters, e.g., $N_s \in [400, |X_{train}|]$. Since we are explaining clusters (and not predicting on unseen data), the training, validation and test sets are identical.

◇ **Experiment Setup**
The comparison is performed over five datasets (limited to 1000 instances), and for each dataset, $k = 2, 3, ..., 10$ clusters are produced. Results for the cost ratio (Equation 12) are reported over *five* trials. Evaluations are performed over the following publicly available datasets: *avila*, *covtype*, *covtype.binary*, *Sensorless* Chang & Lin (2011b) and *mice-protein* Dua & Graff (2017). We *specifically picked* these datasets since CART is known to perform poorly on them Frost et al. (2020); Laber et al. (2021), and thus these provide a good opportunity to showcase the power of this technique.

◇ **Observations**
Figure 7 presents our results. Figure 7(a) shows the plot for the *mice-protein* dataset: the $95\%$ confidence interval, in addition to cost ratio, is shown[13]. Plots for other datasets are shown miniaturized - (b), (c), (d), (e) in the interest of space. The cost for k-means is shown for reference a blue horizontal line at $y = 1$. Figure 7(f) shows the *mean ranks* of the various techniques (lower is better) across datasets and number of clusters (as described in §A.10.1, trials scores are averaged), and its title shows the *p-value*$= 6.688 \times 10^{-6}$ of a *Friedman test* conducted over *the top three techniques*: we restrict the test to top candidates since otherwise it would be very easy to obtain a low score

---

[13]It might come as a surprise that the cost ratio increases with increasing $k$, but this seems to be a transient phenomenon; at even higher values of $k$ we do observe that cost ratios collectively decrease

favorable to us, due to the high cost ratios for CART. The low score indicates with high confidence that ExShallow, IMM and c_CART do not produce the same outcomes.

From the plot of mean ranks in Figure 7(f), we observe that although CART performs quite poorly, the application of our technique drastically improves its performance, to the extent that it competes favorably with techniques like IMM and ExShallow; its mean rank places it between them. This is especially surprising given that it doesn't explicitly minimize the cost in Equation 11. We also note the following *p-values* from *Wilcoxon signed-rank* tests:

- CART vs c_CART: $p = 1.4783 \times 10^{-6}$. The low value indicates that using our technique indeed significantly changes the accuracy of CART.
- IMM vs c_CART: $p = 0.0155$. The relatively high value indicates that the performance of c_CART is competitive with IMM.

Here, both the Friedman and Wilcoxon tests are performed for combinations of datasets and $k$ - a "pseudo-dataset", as discussed in §A.10.1.

### A.10.3    PROTOYPE-BASED CLASSIFICATION

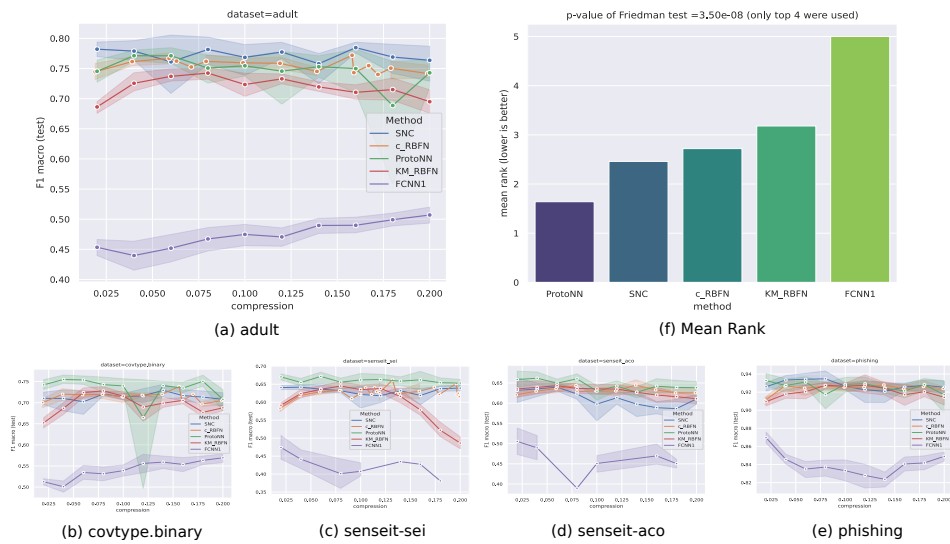

(a) adult                                                      (f) Mean Rank

(b) covtype.binary    (c) senseit-sei    (d) senseit-aco    (e) phishing

Figure 8: Various prototype-based classifiers are compared. (a) shows comparison for the dataset *adult*. Number of prototypes are shown as percentage of the training data on the *x-axis*, and is referred to as "compression". (b), (c), (d) and (e) shows plots for other datasets - these are miniaturized to fit the page. (f) shows the mean ranks of techniques based on five datasets; the Friedman test is conducted over the **top four** techniques only, with $p = 3.5025 \times 10^{-8}$.

.

Next, we consider prototype-based classification. At training time, such techniques identify "prototypes" (actual training instances or generated instances), that maybe used to classify a test instance based on their similarity to them. A popular technique in this family is the *k-Nearest Neighbor (kNN)*. These are simple to interpret, and if a small but effective set of protoypes maybe identified, they can be convenient to deploy on edge devices Gupta et al. (2017); Zhang et al. (2020). Prototypes also serve as minimal "look-alike" examples for explaining models (Li et al., 2018; Nauta et al., 2021). Research in this area has focused on minimizing the number of prototypes that need to be retained while minimally trading off accuracy.

We define some notation first. The number of prototypes we want is an input to our experiments, and is denoted by $N_p$. We will also use $K_\gamma(x_i, x_j) = e^{-\gamma||x_i - x_j||_2^2}$ to denote the *Radial Basis Function (RBF) kernel*, parameterized by the kernel bandwidth $\gamma$.

◇ **Algorithms and Hyperparameters**

These are the algorithms we compare:

1. **ProtoNN** Gupta et al. (2017): This technique uses a RBF kernel to aggregate influence of prototypes. Synthetic prototypes are learned and additionally a "score" is learned for each of them that designates their contribution towards *each* class. The prediction function sums the influence of neighbors using the RBF kernel, weighing contribution towards each class using the learned score values; the class with the highest total score is predicted. The method also allows for reducing dimensionality, but we don't use this aspect[14]. The various parameters are learned via gradient based optimization.

    We use the *EdgeML* library (Dennis et al., 2021), which contains the reference implementation for ProtoNN. For optimization, the implementation uses the version of *ADAM* Kingma & Ba (2015) implemented in *TensorFlow* Abadi et al. (2015); we set $num\_epochs = 200$, $learning\_rate = 0.05$, while using the defaults for other parameters. The $num\_epochs$ and $learning\_rate$ values are picked based on a limited search among values $\{100, 200, 300\}$ and $\{0.01, 0.05\}$ respectively. The search space explored for $\gamma$ is $[0.001, 0.01, 0.1, 1, 10]$. Defaults are used for the other ProtoNN hyperparameters.

2. **Stochastic Neighbor Compression (SNC)** Kusner et al. (2014): This also uses a RBF kernel to aggregate influence of prototypes, but unlike ProtoNN, the prediction is performed via the *1-NN rule*, i.e., prediction uses only the nearest prototype. The technique bootstraps with randomly sampled $N_p$ prototypes (and corresponding labels) from the training data, and then modifies their coordinates for greater accuracy using gradient based optimization; the labels of the prototypes stay unchanged in this process. This is another difference compared to ProtoNN, where in the latter, each prototype contributes to all labels to varying extents. The technique maybe extended to reduce the dimensionality of the data (and prototypes); we don't use this aspect.

    We were unable to locate the reference implementation mentioned in the paper, so we implemented our own version, with the help of the *JAXopt* library Blondel et al. (2021). For optimization, gradient descent with *backtracking line search* is used. A total of $100$ iterations for the gradient search is used (based on a limited search among these values: $\{100, 200, 300\}$), and each backtracking search is allowed up to $50$ iterations. A grid search over the following values of $\gamma$ is performed: $[0.001, 0.01, 0.1, 1, 10]$.

3. **Fast Condensed Nearest Neighbor Rule** Angiulli (2005): Learns a "consistent subset" for the training data: a subset such that for any point in the training set (say with label $l$), the closest point in this subset also has a label $l$. Of the multiple variations of this technique proposed in Angiulli (2005), we use **FCNN1**, which uses the *1-NN* rule for prediction. There are no parameters to tune. We used our own implementation.

    A challenge in benchmarking this technique is it *does not* accept $N_p$ as a parameter; instead it iteratively produces expanding subsets of prototypes until a stopping criteria is met, e.g., if prototype subsets $V_i$ and $V_{i+1}$ are produced at iterations $i$ and $i + 1$ respectively, then they satisfy the relationship $V_i \subset V_{i+1}$. For comparison, we consider the performance at iteration $i$ to be the result of $N_p$ prototypes where $N_p$ is defined to be $|V_i|$, i.e., instead of setting $N_p$, we use the value the algorithm produces at each iteration.

4. **RBFN**: For the traditional model, we use *Radial Basis Function Networks (RBFN)* Broomhead & Lowe (1988). For a binary classification problem with classes $\{-1, 1\}$, given prototypes $x_i, i = 1, 2, ..., p$, the label of a test instance $x$ is predicted as $sgn(\sum_i^p w_i K_\gamma(x, x_i))$ (a score of 0 is set to a label of 1). Weights $w_i$ are learned using linear regression. A one-vs-rest setup is used for multiclass problems. For our baseline, we use cluster centres of a *k-means* clustering as our prototypes, where $k$ is set to $N_p$. These results are denoted using the term **KM_RBFN**. In our version, denoted by **c_RBFN**, the $N_p$ prototypes are sampled from the training data. $N_p$ represents *model size $\eta$* here.

    Note that the standard RBFN, and therefore the variants used here KM_RBFN and c_RBFN, don't provide a way to reduce dimensionality; this is the reason why this aspect of ProtoNN and SNC wasn't used (for fair comparison). We set $T = 1000$ and $N_s$ was set to $N_p$ to get the desired number of prototypes.

---

[14]The implementation provides no way to switch off learning a projection, so we set the dimensionality of the projection to be equal to the original number of dimensions. This setting might however learn a transformation of the data to space within the same number of dimensions, e.g., translation, rotation.

Although all the above techniques use prototypes for classification, it is interesting to note variations in their design: ProtoNN, SNC, KM_RBFN use synthetic prototypes, i.e., they are not part of the training data, while c_RBFN and FCNN1 select $N_p$ instances from the training data. The prediction logic also differs: ProtoNN, KM_RBFN, c_RBFN derive a label from some function of the influence by all prototypes, while SNC and FCNN1 use the 1-NN rule.

⋄ **Experiment Setup**

As before, we evaluate these techniques over five standard datasets: *adult, covtype.binary, senseit-sei, senseit-aco, phishing* Chang & Lin (2011b). 1000 training points are used, with $N_p \in \{20, 40, 60, 80, 100, 140, 160, 180, 200\}$. Results are reported over five trials. The score reported is the F1-macro score.

⋄ **Observations**

Results are shown in Figure 8. (a) shows the plot for the *adult* dataset. The number of prototypes are shown on the *x-axis* as *percentages* of the training data. Plots for other datasets are shown in (b), (c), (d) and (e); these have been miniaturized to fit the page. Figure 8(f) shows the mean rank (lower is better) across datasets and number of prototypes (as described in §A.10.1, trials are aggregated over). The p-value of the Friedman test is reported, $p = 3.5025 \times 10^{-8}$. Here too, we do not consider the worst performing candidate, FCNN1 - so as to not bias the Friedman test in our favor.

We observe in Figure 8(f) that while both ProtoNN and SNC outperform c_RBFN, the performance of SNC and c_RBFN are close. We also observe that FCNN1 performs poorly; this matches the observations in Kusner et al. (2014).

We also consider the following *p-values* from *Wilcoxon signed-rank* tests:

1. KM_RBFN vs c_RBFN: $p = 1.699 \times 10^{-4}$. The low value indicates that our technique significantly improves upon the baseline KM_RBFN.

2. SNC vs c_RBFN: $p = 0.1260$. The relatively high value here indicates that c_RBFN is competitive with SNC; in fact, at a confidence threshold of $0.1$, their outcomes would not be interpreted as significantly different.

As discussed in §A.10.1, these statistical tests are conducted over a combination of dataset and model size.

### A.11 RUNTIMES

For our experiments in the main paper, we used the *hyperopt* library on account of its popularity and maturity. Its acquisition function approximates the *Probability of Improvement (PI)* utility function (Song et al., 2022), which can exhibit greedy behavior (Garnett, 2023). In contrast, we might use a different utility function, such as *Expected Improvement (EI)*, which is relatively more exploratory, and thus, is likely to find better extrema.

We present some initial results around this line of thought. Instead of using the naive EI, we use a numerically stable version called *LogEI* (Ament et al., 2023) from the *BoTorch* (Balandat et al., 2020) package. We also note that we might use an acquisition function that can explicitly account for noise[15], thus bypassing the need for estimating $s_t$ in Algorithm 3 via averaging (see notes in §A.4).

Table 6 shows results for datasets *a1a* and *ijcnn1*, where the interpretable model is a DT and the oracle is a GBM. We assume *homoscedastic* noise with variance of $0.5$. *hyperopt* was provided a budget of $T = 3000$ evaluations as in the main paper, while *BoTorch* was allowed $T = 200$ iterations. We observe that significant speedups are obtained without mostly noticeable change in the quality of results - the only exception seems to be for *ijcnn1*, for $depth = 2$. To take an example, for the dataset *a1a*, for DT $depth = 1$, the time taken by *hyperopt* is 3193.27 seconds or **53 minutes**, while *BoTorch* offers of speedup of 21.57x; this is a runtime of $3193.27/21.57 = 148.04$ seconds or $\sim$ **2 minutes**.

---

[15]We use this particular function: `https://botorch.readthedocs.io/en/latest/acquisition.html#botorch.acquisition.analytic.LogNoisyExpectedImprovement`.

Table 6: Difference between using *BoTorch* with the noisy *LogEI* acquisition function, and *hyperopt*. The table shows: (a)*hyperopt* runtimes (in seconds), (b) percentage point (pp) difference between the %-age improvements seen between *BoTorch* and *hyperopt*, and (c) the speedup in wallclock runtime with *BoTorch*. *BoTorch* and *hyperopt* were run for $200$ and $3000$ iterations respectively. For these examples, *BoTorch* runs significantly faster. Aside from one case - *ijcnn1*, $depth = 2$,the performance degradation is reasonable. In some cases, it seems to perform better. **Results are averaged over three runs.**

| dataset | tree depth $= 1$ | 2 | 3 | 4 | 5 |
|---------|------------------|---|---|---|---|
| a1a | $\text{time}_{\text{hyp}} = 3193.27\text{s},$ $\text{pp} = +0.19,$ $\text{speedup} = 21.57\text{x}$ | 4228.81s, $+1.87$ 27.02x | 3867.64s, $+3.22,$ 25.23x | 4843.54s, $-0.43,$ 34.50x | 3610.96s, $+4.04,$ 27.86x |
| ijcnn1 | 4221.34s, $-2.39,$ 28.37x | 3902.53s, $-7.84$ 24.62x | 4613.63s, $-3.47,$ 29.01x | 4362.36s, $-1.36,$ 27.20x | 4962.23s, $+0.20,$ 29.59x |

## A.12 MULTIVARIATE MODEL SIZES

Our technique is applicable even when the model size has more than one attribute. This is because Algorithm 1 delegates size enforcement to $train_{\mathcal{I},g}$. Consider GBMs, where we might consider a bivariate size, $\eta = [max\_depth, num\_trees]$; here the quantities respectively denote the maximum depth allowed for each constituent DT in a GBM, and the number of DTs in the GBM. In Figure 9, we show how improvements for GBMs vary when $1 \leq max\_depth \leq 5$ ($x$-axis) and $1 \leq num\_trees \leq 5$ ($y$-axis); the oracle used is a GBM as well (unconstrained in size). Results are averaged over *three* runs for these datasets: (a) *senseit-sei* (b) *higgs* (c) *cod-rna* and (d) *senseit-aco* here. We continue to observe pattern that as model sizes increase, in terms of both $max\_depth$ and $num\_trees$, improvements decrease.

## A.13 DIFFERENT FEATURE SPACES

In our validation experiments in §4.1.2, the feature vector representation was identical for the oracle and the interpretable model. This is also what Algorithm 1 implicitly assumes. Here, we consider the possibility of going a step further and using different feature vectors. If $f_{\mathcal{O}}$ and $f_{\mathcal{I}}$ are the feature vector creation functions for the oracle and the interpretable model respectively, and $x_i$ is a "raw data" instance, then:

1. The oracle is trained on instances $f_{\mathcal{O}}(x_i)$, and provides uncertainties $u_{\mathcal{O}}(f_{\mathcal{O}}(x_i))$.

2. The interpretable model is provided with data $f_{\mathcal{I}}(x_i)$, but the uncertainty scores available to it are $u_{\mathcal{O}}(f_{\mathcal{O}}(x_i))$.

The motivation for using different feature spaces is that the combination $(\mathcal{O}, f_{\mathcal{O}})$ may be known to work well together and/or a pre-trained oracle might be available only for this combination.

We illustrate this application with the example of predicting nationalities from surnames of individuals. Our dataset (Rao & McMahan, 2019) contains examples from $18$ nationalities: *Arabic, Chinese, Czech, Dutch, English, French, German, Greek, Irish, Italian, Japanese, Korean, Polish, Portuguese, Russian, Scottish, Spanish, Vietnamese*. The representations and models are as follows:

1. The oracle model is a *Gated Recurrent Unit (GRU)* (Cho et al., 2014), that is learned on the sequence of characters in a surname. The GRU is calibrated with *temperature scaling* (Guo et al., 2017).

2. The interpretable model is a DT, where the features are character n-grams, $n \in 1, 2, 3$. The entire training set is initially scanned to construct an n-gram vocabulary, which is then used to create a sparse binary vector per surname - 1s and 0s indicating the presence and absence of an n-gram respectively.

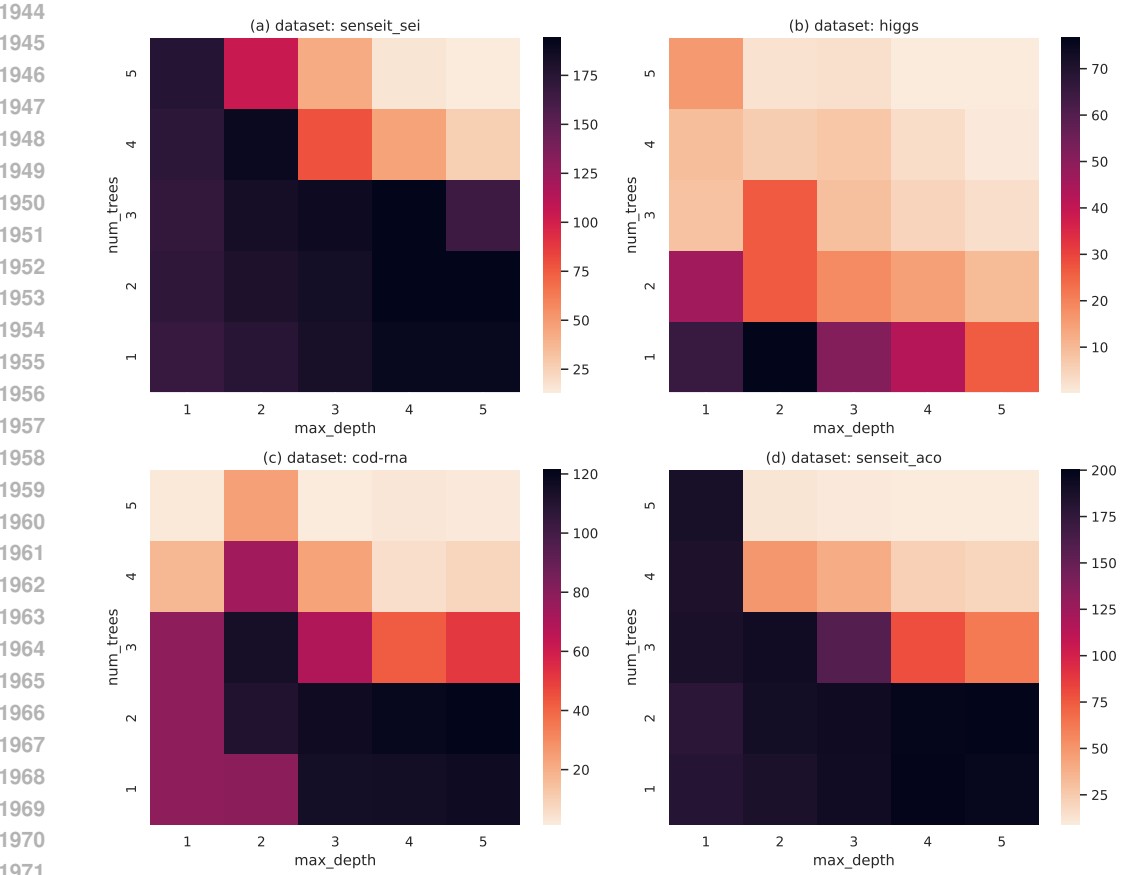

Figure 9: Improvements in test $F1$-macro for multiple datasets for different sizes of $GBM$ models are shown. (a) Top-left: *senseit-sei* (b) Top-right: *higgs* (c) Bottom-left:*cod-rna* and (d) Bottom-right: *senseit-aco*. Here, model size is the combination of *max_depth* and *number of trees* in the $GBM$ model. Greater improvements are seen at lower sizes.

Figure 10 shows a schematic of the setup.

The n-gram representation leads to a vocabulary of $\sim 5000$ terms, that is reduced to 600 terms based on a $\chi^2$-test in the interest of lower running time. DTs of different $depth \leq 15$ were trained. A budget of $T = 3000$ iterations was used, and the relative improvement in the $F1$ macro score (as in Equation 3) is reported, averaged over three runs. Figure 11 shows the results.

We see large improvements at small depths, that peak with $\delta F1 = 83.04\%$ at $depth = 3$, and then again at slightly larger depths, which peak at $depth = 9$ with $\delta F1 = 12.34\%$.

To obtain a qualitative idea of the changes in the DT using a oracle produces, we look at the prediction rules for *Polish* surnames, when DT $depth = 3$. For each rule, we also present examples of true and false positives.

**Baseline rules** - $precision = 2.99\%, recall = 85.71\%, F1 = 5.77\%$:

Rule 1. $k \wedge ski \wedge \neg v$

- True Positives: *jaskolski, rudawski*
- False Positives: *skipper (English), babutski (Russian)*

Rule 2. $k \wedge \neg ski \wedge \neg v$

- True Positives: *wawrzaszek, koziol*
- False Positives: *konda (Japanese), jagujinsky (Russian)*

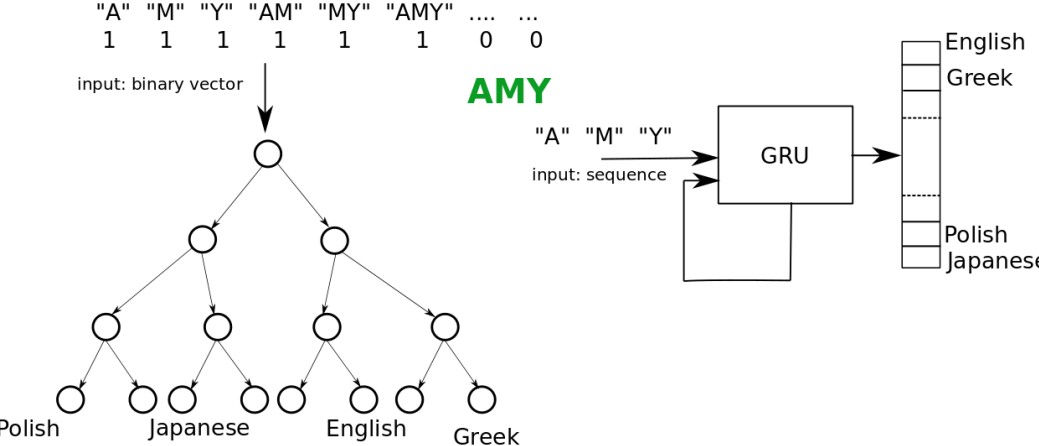

Figure 10: The feature representations for the oracle and the interpretable model may be different. Consider the name "Amy": the GRU is provided its letters, one at a time, in sequence, while the DT is given an n-gram representation of the name.

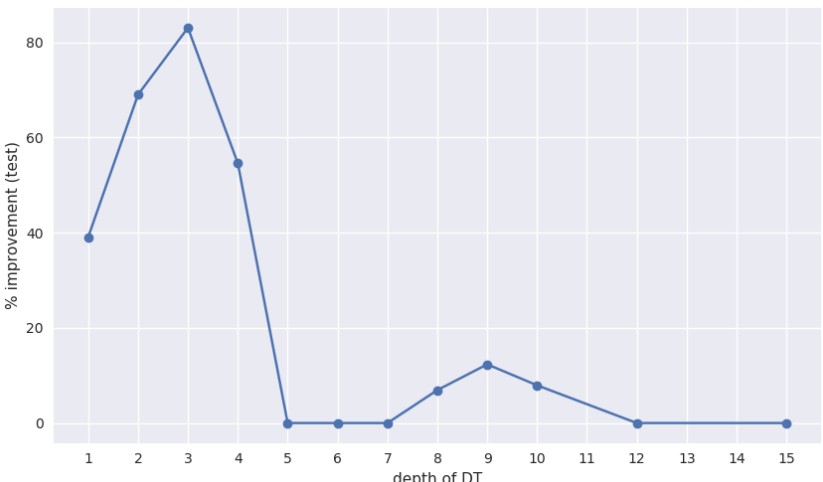

Figure 11: Improvements $\delta F1$ are shown for different depths of the DT.

**Oracle-based DT rules** - $precision = 25.00\%, recall = 21.43\%, F1 = 23.08\%$:

Rule 1. $ski \wedge \neg(b \vee kin)$

- True Positives: *jaskolski, rudawski*
- False Positives: *skipper (English), aivazovski (Russian)*

We note that the baseline rules are in conflict w.r.t. the literal "ski", and taken together, they simplify to $k \wedge \neg v$. This makes them extremely permissive, especially *Rule 2*, which requires the literal "k" while needing "ski" and "v" to be absent. Not surprisingly, these rules have high recall ($= 85.71\%$) but poor precision ($= 2.99\%$), leading to $F1 = 5.77\%$.

In the case of the oracle-based DT, now we have only one rule, that requires the atypical trigram "ski". This improves precision ($= 25\%$), trading off recall ($= 21.43\%$), for a significantly improved $F1 = 23.08\%$.

The difference in rules may also be visualized by comparing the distribution of nationalities represented in their false positives, as in Figure 12. We see that the baseline DT rules, especially *Rule 2*, predict many nationalities, but in the case of the DT learned using the oracle, the model confusion

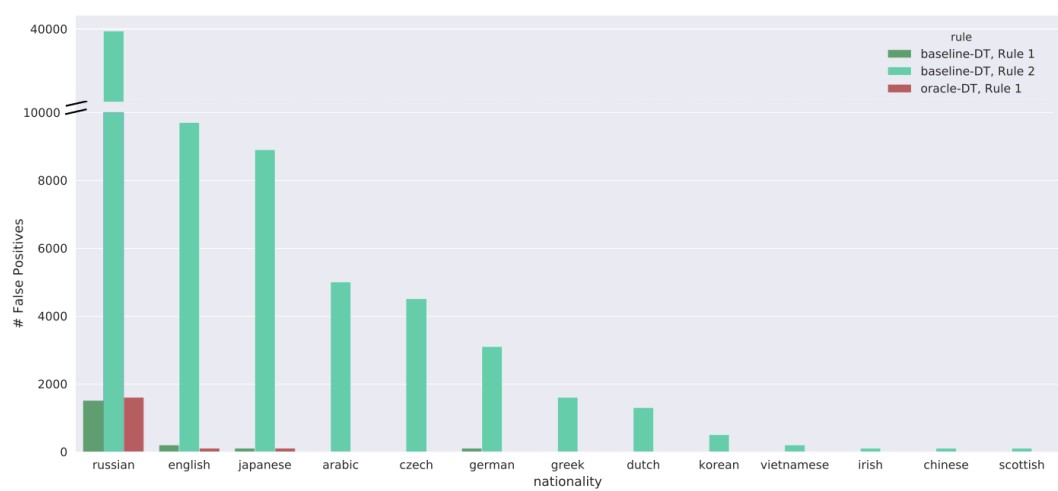

Figure 12: The distribution of nationalities in false positive predictions for the baseline and oracle based models, shown for predicting *Polish* names. Only nationalities with non-zero counts are shown.

is concentrated around *Russian* names, which is reasonable given the shared *Slavic* origin of many *Polish* and *Russian* names.

We believe this is a particularly powerful and exciting application of our technique, and opens up a wide range of possibilities for translating information between models of varied capabilities.

