# OpenReview forum: "Learning Interpretable Models Using Uncertainty Oracles"
_ICLR.cc/2026/Conference — Submitted to ICLR 2026_

### Official Review · Reviewer_6JER · 2025-10-26

**Soundness:** 2
**Presentation:** 2
**Contribution:** 2
**Rating:** 6
**Confidence:** 4

**Summary:**

The paper presents a framework for constructing small, interpretable models that achieve high accuracy by leveraging the training distribution through a flexible probabilistic modeling approach. The core idea is to use a Dirichlet Process (DP) to model the training data distribution, enabling adaptive and expressive representation learning that supports small model sizes without sacrificing performance. The DP’s parameters are optimized via Bayesian Optimization, a design choice that allows the method to handle non-differentiable loss functions, which is critical for many real-world interpretability tasks. To manage high-dimensional input spaces, the method employs data projection based on uncertainty scores from a separate probabilistic model. This oracle can be any well-performing model, and its predictions guide the dimensionality reduction, enabling the framework to work across diverse feature spaces. The authors demonstrate that their approach improves the accuracy of small interpretable models across multiple model families. It is modular and general: it can be applied to various model types, different feature representations, and even enhance older, classic algorithms to perform competitively with modern, task-specific methods on tasks such as cluster explanation.

**Strengths:**

1. The proposed method achieves improvements in accuracy for small, interpretable models addressing a core challenge in the field: the severe accuracy penalty incurred when enforcing model simplicity for human interpretability.
2. The technique is not tied to a specific model class or size metric. It can be applied to a wide range of interpretable models and supports diverse notions of "size," including depth, number of non-zero coefficients, and combined constraints. This versatility significantly broadens its applicability.
3. The method requires only one hyperparameter, the optimization budget, which is a major practical advantage over competing approaches that require tuning multiple, complex hyperparameters. This simplicity enhances usability and reproducibility.
4. By leveraging Bayesian Optimization to tune the Dirichlet Process parameters, the method is inherently suitable for non-differentiable loss functions, a critical feature that enables application to discrete, combinatorial model spaces. This is a notable advantage over gradient-based optimization methods, which fail in such settings.
5. The framework permits the use of an "uncertainty oracle" to guide the training of an interpretable model. This decoupling of the oracle and model feature spaces enables powerful transfer of uncertainty information across heterogeneous representations, enhancing generalization and flexibility.
6. Experiments cover both binary/multi‑class classification, and compare against multiple baselines.

**Weaknesses:**

1. The reliance on Bayesian Optimization leads to prohibitively long training times. While the authors acknowledge this and suggest alternative optimization strategies, the current implementation remains computationally expensive, limiting scalability to large datasets or real-time applications.
2. Although the intuition behind learning the training distribution is plausible, the paper offers no theoretical analysis to explain why this approach yields such large improvements in accuracy. The absence of a formal argument weakens the manuscript and makes it difficult to predict under what conditions the method may fail.
3. The method’s performance hinges on the quality of the “uncertainty oracle.” However, the paper neither examines how different oracle choices influence results nor quantifies the model’s sensitivity to oracle uncertainty estimates. Potential error propagation from a faulty oracle is unaddressed. Moreover, if the oracle is miscalibrated or exhibits systematic bias, the guided model may inherit these issues. The paper also omits an analysis of robustness to oracle noise.
4. The oracle must be trained beforehand on the full dataset, which can be expensive for large‑scale problems. The paper does not quantify the total runtime overhead relative to training a standalone tree.
5. The optimization budget (3000 iterations) is relatively high for a single depth level. The paper does not discuss the sensitivity to this budget.
6. It would also be valuable to assess the technique on more complex real-world applications beyond the benchmark datasets to better demonstrate the practical applicability and performance of the proposed method.

**Questions:**

Please refer to the section on weaknesses.

---

### Official Review · Reviewer_Ez5o · 2025-10-28

**Soundness:** 2
**Presentation:** 3
**Contribution:** 2
**Rating:** 4
**Confidence:** 3

**Summary:**

The paper proposes a two-stage framework: first, an "uncertainty oracle" scores samples, and a Dirichlet Process-driven Infinite Beta Mixture Model is used to learn a target sampling distribution on the uncertainty axis, thereby resampling the training set; subsequently, a small interpretable model is trained on the resampled data. The goal is to improve the interpretability and generalization performance of the final model through "explanation-oriented and robust" data selection.

**Strengths:**

The method is model-agnostic and applicable to various interpretable models.

**Weaknesses:**

1. While Step 2 primarily uses small models like decision trees and linear models, which is acceptable given the focus on surrogate models for interpretability, the datasets used in the experiments are also very small, raising doubts about the method's practical utility.

2. Furthermore, the runtime issue highlighted by the authors in the limitations section seems critical. With runtimes already approaching an hour even for small datasets, the time sacrifice may not be justified by the performance gains achieved.

**Questions:**

1. Since the oracle and the interpretable model are decoupled, their decision boundaries may not align. How can it be ensured that the decision boundary of the model in Step 1 is the same as or congruent with that in Step 2?


2. The framework's effectiveness heavily depends on whether the oracle's uncertainty scores are truly calibrated and reflect the "difficulty relative to the decision boundary." The paper lacks experimental validation of the uncertainty estimation.

---

### Official Review · Reviewer_CnJ6 · 2025-11-05

**Soundness:** 2
**Presentation:** 2
**Contribution:** 1
**Rating:** 0
**Confidence:** 4

**Summary:**

In this paper, a training technique is proposed  to improve small interpretable models accuracy by learning training distribution of uncertainty scores from an oracle model. They iteratively train a Dirichlet Process based mixture model to represent the distribution over 1D-projected uncertainty scores through Bayesian Optimization and target model F1 score. The method is designed to be practically versatile, handling non-differentiable loss functions, cross-feature-space and multivariate model sizes. With thorough empirical experiment, the method demonstrates sound and board improvements across 13 standard classification datasets on a variety of interpretable model families, clearly highlighted with Wilcoxon signed-rank tests.

**Strengths:**

None

**Weaknesses:**

The contribution is unclear due to lack of theory grounding, computational costs disadvantage, less technical novelty and issues on methodology.
- Lack of theory grounding - The paper covers very little theories, explanation and interpretations of experiment findings. There are no clear discussions or theoretical justifications on why uncertainty-based sampling improves small models. Formal analysis of convergence properties is also missed in the paper.
- Unfavorable computational cost and unstudied complexity - The high iterations and runtime overhead for small model training defeats the purpose for practical interpretable ML applications. In  addition, runtime comparisons with standard interpretable ML training could be helpful to understand the limitations of this technique.
- Less technical novelty - The core methodology is a specific combination of well-understood and standard techniques, lacking fundamental innovations.
- Methodological Issues - The method has strong oracle dependence, assuming oracle quality is well constructed and calibrated. There is no discussion on robustness to oracle degradation or choices. Secondly, although the paper provides a suggested default reasonable settings, the "only one hyperparameter" is not factual. A variety of hyperparameters are involved in methodology, including uncertainty metric choice, smoothing parameters, box constraints and oracle architecture.

**Questions:**

Interpretation on negative improvements: Is there any detailed analysis and interpretation of negative improvements in Table 4?

---

### Meta-Review · Area_Chair_9e7P · 2026-01-07

**Summary:**

Small, interpretable models are easier for humans but often suffer accuracy loss. Standard size-control levers (such as depth limits, L1, early stopping) may not reach a desirable size–accuracy trade-off. Instead of changing the model, learn a better training distribution. Instances are reweighted/sampled so that a size-constrained model trained on this distribution achieves higher accuracy on the original test distribution.

**Reviewer Concerns:**

The reveiwers raised several concerns:

CnJ6 -- Strongly rejected the paper citing concerns for lack of theory / explanation, Computational Cost & Practicality (includig missing run time comparisons), lack of technical novelty,  and some methodological issues such as strong oracle dependency and misrepresentation in how many hyper-parameters are involved.

Ez5o -- Said that the idea is promising but had some practicality concerns like limited experiments + scale of datasets used, longer runtime, possible mislalignment and micalibration with the oracle

6JER-- also schoed CnJ6's concern about lack of theory/explanation, and Ez5o's concern about runtime, sensitivity to the oracle and sensitivity to the oracle.

There was no rebuttal and so none of the above concerns were addressed.

**Reviewer Scores:**

Since there was no rebuttal I do not forsee any change in the rebuttal scores.

---

### Decision · Program_Chairs · 2026-01-26

Reject